# An Offline Adaptation Framework for Constrained Multi-Objective Reinforcement Learning

**Qian Lin[1], Zongkai Liu[1], Danying Mo[1], Chao Yu[1,2,3†]**
[1]Sun Yat-sen University, Guangzhou, China
[2]Pengcheng Laboratory, Shenzhen, China
[3]MoE Key Laboratory of Information Technology, Guangzhou, China
`{linq67,liuzk,mody3}@mail2.sysu.edu.cn, yuchao3@mail.sysu.edu.cn`

## Abstract

In recent years, significant progress has been made in multi-objective reinforcement learning (RL) research, which aims to balance multiple objectives by incorporating preferences for each objective. In most existing studies, specific preferences must be provided during deployment to indicate the desired policies explicitly. However, designing these preferences depends heavily on human prior knowledge, which is typically obtained through extensive observation of high-performing demonstrations with expected behaviors. In this work, we propose a simple yet effective offline adaptation framework for multi-objective RL problems without assuming handcrafted target preferences, but only given several demonstrations to implicitly indicate the preferences of expected policies. Additionally, we demonstrate that our framework can naturally be extended to meet constraints on safety-critical objectives by utilizing safe demonstrations, even when the safety thresholds are unknown. Empirical results on offline multi-objective and safe tasks demonstrate the capability of our framework to infer policies that align with real preferences while meeting the constraints implied by the provided demonstrations.

## 1 Introduction

In the standard reinforcement learning (RL) setting, the primary goal is to obtain a policy that maximizes a cumulative scalar reward [Sutton and Barto, 2018]. However, in many real-world applications involving multiple objectives, a desired policy must not only strike a balance among potentially conflicting objectives but also consider the constraints imposed on specific safety-critical objectives. Such requirements motivate the research of Multi-objective RL (MORL) [Liu et al., 2014, Mossalam et al., 2016] and safe RL [Gu et al., 2022, Achiam et al., 2017]. In addition to reward maximization, the former aims to enable policies that cater to a target preference indicating the trade-off between competing objectives, while the latter focuses on reducing the cost measures of learned policies within a given safety threshold.

Despite the development of meticulous and effective mechanisms to achieve these goals, most existing MORL and safe RL algorithms rely on predefined target preferences or safety thresholds. These elements need to be carefully designed by human experts with prior knowledge, generalized from a large number of demonstrations through observation. Taking autonomous driving as an example, defining aggressive, stable, and conservative strategies through different preferences between minimizing energy consumption and increasing driving speed may require extensive human participation, which means that researchers need to categorize driving data into different styles based on human experience and observe the energy consumption-speed ratio to estimate appropriate preference weights and

---

[†]Corresponding author

38th Conference on Neural Information Processing Systems (NeurIPS 2024).

safety thresholds for different strategies. Moreover, it is uncertain whether policies based on manually designed preferences will exhibit expected behaviors or whether feasible policies exist under given safety constraints. The challenge of designing appropriate preferences or safety thresholds becomes more pronounced as the number of objectives increases.

Compared to manually designing target preferences or safety thresholds based on human knowledge, it is more natural to infer expected behaviors through a few demonstrations that implicitly indicate the trade-off between multiple objectives and the constraints of safety. For instance, selecting demonstrations with conservative behaviors from the driving dataset can be easier than inferring preferences that lead to conservative policies. Therefore, in this work, we formulate a novel offline adaptation problem for constrained MORL, with the goal to leverage a few demonstrations with expected behaviors, rather than relying on handcrafted target preferences and safety thresholds, to generate a target policy that achieves desired trade-offs across various objectives and meets the constraints on safety-critical objectives under offline settings.

To achieve this, we first focus on unconstrained MORL scenarios and propose a simple yet effective offline adaptation framework **Preference Distribution Offline Adaptation (PDOA)**, which includes: 1) learning a set of policies that respond to various preferences during training, and 2) adapting a distribution of target preferences based on a few given demonstrations during deployment. Specifically, we initialize the first part with existing state-of-the-art offline MORL algorithms, and then in the second part, we propose to align the adapted policies with expected behaviors by modeling the posterior preference distribution regarding demonstrations. Moreover, we show that our framework can be extended to constrained MORL settings by converting a constrained RL problem into an unconstrained MORL counterpart, and incorporating a conservative estimate of preference weights on constrained objectives to mitigate the potential constraint violations. Lastly, we conduct several empirical experiments on classical MORL, safe RL tasks and a novel constrained MORL environment under offline settings, demonstrating the capability of our framework in generating policies that align with the real preferences and meet the constraints implied in demonstrations.

## 2 Related Work

**Multi-objective RL**  Many existing MORL methods explicitly maintain a set of policies tailored to various given preferences to approximate the Pareto front of optimal policies. These works either apply a single-policy algorithm individually for each candidate preference [Roijers et al., 2014, Mossalam et al., 2016], employ evolutionary algorithms to generate a population of diverse policies [Handa, 2009, Xu et al., 2020] or simultaneously learn a set of policies represented by a single network [Abels et al., 2019, Basaklar et al., 2022]. Furthermore, due to the potential costs and risks associated with extensive online exploration, several studies [Zhu et al., 2023, Lin et al., 2024] have been proposed to leverage only offline datasets for MORL by extending offline return-conditioned methods [Chen et al., 2021, Emmons et al., 2021] or offline policy-regularized methods [Fujimoto and Gu, 2021, Wang et al., 2022] to MORL settings. Despite the ability to obtain a set of well-performing policies for various preferences, most existing MORL methods overlook the process of acquiring target preferences for identifying appropriate policies during practical deployment. In this work, we assume no online interactions and no target preferences but several demonstrations generated with expected behaviors, which are easier to access than meticulously designed target preferences.

**Safe RL**  While maximizing the expected reward, classical safe RL methods restrict the cumulative cost to stay within a predefined safety threshold through Lagrangian primal-dual methods [Stooke et al., 2020, Chow et al., 2017] or primal approaches [Xu et al., 2021, Sootla et al., 2022]. Recently, several studies have focused on learning a set of policies that respond to various safety thresholds in both online settings [Yao et al., 2024a] and offline settings [Liu et al., 2023a, Lin et al., 2023]. Similar to MORL, these works also assume access to well-designed safety thresholds that ensure safe behaviors. Unlike prior research, our framework relies solely on safe demonstrations to indicate the implicit constraints and offers a mechanism to mitigate constraint violations through conservatism.

**RL with Offline Adaptation**  Several studies have applied meta-learning techniques to address MORL [Chen et al., 2019] or safe RL problems [Guan et al., 2024a], but they require online interactions for task adaptation and suffer from low sample efficiency. Additionally, other research endeavors explore offline adaptation methods to mitigate environmental uncertainty by modelling a posterior distribution over all possible Markov Decision Processes (MDPs) [Ghosh et al., 2022] or

considering varying confidence levels in conservative value estimates [Hong et al., 2022]. [Mitchell et al., 2021, Xu et al., 2022a] focus on offline adaptation for multi-task problems, aiming to achieve fast adaptation to new downstream tasks after offline training on multi-task experience. Among these methods, offline meta RL [Mitchell et al., 2021] addresses a problem similar to ours, which aims to train a meta policy that can adapt to a new task with limited data. Prompt-DT [Xu et al., 2022a] achieves quick adaptation to new tasks by incorporating a few demonstrations as prompts into the decision transformer [Chen et al., 2021] framework. We present further discussions about the difference between multi-task RL and our setting in Appendix A.1.

## 3 Preliminaries

### 3.1 Constrained Multi-Objective MDP (CMO-MDP)

Both multiple-objective RL and safe RL can be discussed based on a uniform formulation: constrained multi-objective MDP (CMO-MDP) proposed by LP3 [Huang et al., 2022]. A CMO-MDP is defined as a tuple $(\mathcal{S}, \mathcal{A}, \mathcal{P}, \boldsymbol{r}, \boldsymbol{c}, \boldsymbol{\beta}, \gamma)$ with state space $\mathcal{S}$, action space $\mathcal{A}$, transition distribution $\mathcal{P}(s'|s, a)$, vector reward functions $\boldsymbol{r} \in R^N$ for $N$ unconstrained objectives, vector cost functions $\boldsymbol{c} \in R_+^K$, safety thresholds $\boldsymbol{\beta} \in R_+^K$ for $K$ constrained objectives and discount factor $\gamma \in [0, 1]$. The goal is to maximize the rewards on unconstrained objectives while ensuring the costs on constrained objectives remain within the safety threshold $\boldsymbol{\beta}$. Since it is typically infeasible to maximize all task objectives simultaneously, preferences $\boldsymbol{\omega} \in \Omega$ and preference functions $f_{\boldsymbol{\omega}}(\boldsymbol{r})$ which map the reward $\boldsymbol{r}$ to a scalar utility under a specific preference $\boldsymbol{\omega}$, are introduced to control the trade-off between unconstrained objectives. Given preferences $\boldsymbol{\omega} \in \Omega$ and safety thresholds $\boldsymbol{\beta}$, the goal can be formulated as follows:

$$\max_{\pi_{\boldsymbol{\omega}, \boldsymbol{\beta}}} \mathbb{E}_{\pi_{\boldsymbol{\omega}, \boldsymbol{\beta}}} \left[ R_{\boldsymbol{\omega}} \right], \text{s.t.} \quad \boldsymbol{C} \preceq \boldsymbol{\beta}, \tag{1}$$

where $R_{\boldsymbol{\omega}} = \sum_t f_{\boldsymbol{\omega}}(\boldsymbol{r}_t)$ and $\boldsymbol{C} = \sum_t \boldsymbol{c}_t$ represent cumulative utility and vector cost over time $t$, respectively. We denote $\pi_{\boldsymbol{\omega}, \boldsymbol{\beta}}$ as a policy conditioned on $\boldsymbol{\omega}, \boldsymbol{\beta}$. In this paper, we consider the linear preference setting (i.e., $f_{\boldsymbol{\omega}}(\boldsymbol{r}) = \boldsymbol{\omega}^\mathsf{T} \boldsymbol{r}$ where $\boldsymbol{\omega} \in R^N$ and $\|\boldsymbol{\omega}\|_1 = 1$), which is widely studied and applied [Mossalam et al., 2016, Abels et al., 2019] and also serves as a bridge between unconstrained and constrained MORL, as shown in Section 4.3. A CMO-MDP problem can degenerate to a standard safe RL problem when $N = 1$ and to a standard multi-objective problem when $K = 0$.

### 3.2 Offline MORL

Under offline MORL settings, an offline dataset $\mathcal{D} = \{(s, a, s', \boldsymbol{r}, \boldsymbol{c}, \boldsymbol{\omega})\}$ is the only data available for training, which is generated by a set of behavior policies $\pi_b(\cdot|\boldsymbol{\omega})$ with diverse behavioral preferences $\boldsymbol{\omega}$. One straightforward yet effective approach to learn a set of policies for various preferences is to adapt offline single-objective RL methods for MORL settings. An example of this approach is the **multi-objective version of Diffusion-QL (MODF)** [Lin et al., 2024], which incorporates a preference-conditioned policy (i.e., $\pi(a|s, \boldsymbol{\omega})$) and a multi-dimensional value function for $N$ objectives (i.e., $\boldsymbol{Q}(s, a, \boldsymbol{\omega}) = Q_1(s, a, \boldsymbol{\omega}), ..., Q_N(s, a, \boldsymbol{\omega})$) into Diffusion-QL [Wang et al., 2022]:

$$\begin{aligned} L_\pi = -\mathbb{E}_{(s, a, \boldsymbol{\omega}) \sim \mathcal{D}} \big[ \mathbb{E}_{a' \sim \pi(\cdot|s, \boldsymbol{\omega})} \left[ \boldsymbol{\omega}^\mathsf{T} \boldsymbol{Q}(s, a', \boldsymbol{\omega}) \right] - \\ \kappa \mathbb{E}_{i \sim \mathcal{U}, \epsilon \sim \mathcal{N}(\boldsymbol{0}, \boldsymbol{I})} \left[ \|\epsilon - \epsilon_\theta(\sqrt{\overline{\alpha}_i} a + \sqrt{1 - \overline{\alpha}_i} \epsilon, s, i)\|^2 \right] \big], \\ L_{\boldsymbol{Q}} = \mathbb{E}_{(s, a, \boldsymbol{r}, s', \boldsymbol{\omega}) \sim \mathcal{D}} \left[ (\boldsymbol{r} + \gamma \mathbb{E}_{a' \sim \pi(\cdot|s, \boldsymbol{\omega})} \boldsymbol{Q}(s', a', \boldsymbol{\omega}) - \boldsymbol{Q}(s, a, \boldsymbol{\omega}))^2 \right], \end{aligned} \tag{2}$$

where $i$ is the diffusion timestep, $\kappa$ is the regularization weight, $\overline{\alpha}_i$ are pre-defined parameters of diffusion model and $\epsilon_\theta(\cdot)$ is a denoiser model. The diffusion policy generates the actions by iteratively using $\epsilon_\theta(\cdot)$ to recover actions from noise. The second term in the actor loss of Eq. (2) is a diffusion reconstruction loss, which serves as a regularization term to align the actions of diffusion policy with the behavioral actions in the dataset.

**Pareto-Efficient Decision Agents (PEDA)** [Zhu et al., 2023] is another MORL method based on supervised RL, which trains a policy conditioned on both target preferences and vector returns through a supervised paradigm:

$$L_\pi = -\mathbb{E}_{(\tau_{t:T}, \boldsymbol{\omega}) \sim \mathcal{D}} \left[ \log \pi(a_t|s_t, \boldsymbol{g}_t, \boldsymbol{\omega}) \right], \tag{3}$$

where $\tau_{t:T} = \{(s_t, a_t, \boldsymbol{r}_t), ..., (s_T, a_T, \boldsymbol{r}_T)\}$ is a trajectory segment, $\boldsymbol{g}_t = \sum_{t'=t}^{T} \boldsymbol{r}_{t'}$ is the target vector return (a.k.a., return-to-go) and $\boldsymbol{\omega}$ is the behavioral preference of $\tau_{t:T}$. It is worth noting that despite the significant performance of the above two methods in offline MORL problems, both require human-provided target preferences during deployment to achieve the desired behavior, and therefore cannot be directly applied to the setting in this paper.

# 4 An Offline Adaptation Framework for Constrained Multi-Objective RL

## 4.1 Problem Formulation of Offline Adaptation for CMO-MDP

In this paper, we focus on a novel offline adaptation problem for constrained MORL, with the goal to leverage only a few demonstrations to generate the policies that exhibit expected behaviors. During training, an offline dataset $\mathcal{D} = \{(s, a, s', \boldsymbol{r}, \boldsymbol{c}, \boldsymbol{\omega})\}$ is provided for policy training. During deployment, we have access to a demonstration set corresponding to a target $G$, i.e.,

$$\mathcal{B}_G = \{x_i \sim \pi_G^*\}_{i=1}^{M}, \tag{4}$$

where $x_i$ is defined as a tuple $(s_i, a_i, s_i', \boldsymbol{r}_i, \boldsymbol{c}_i)$ and $M$ is the total number of transition demonstrations. The target $G$ can be preferences in MORL problems ($G = \boldsymbol{\omega}_g$), safety thresholds in safe problems ($G = \boldsymbol{\beta}_g$) or a combination of both ($G = (\boldsymbol{\omega}_g, \boldsymbol{\beta}_g)$), and $\pi_G^*$ is the expert policy that achieves the best utility and meet the constraints under the preference $\boldsymbol{\omega}_g$ and safety threshold $\boldsymbol{\beta}_g$ of the target $G$. In our setting, the real target $G$ is inaccessible, and the goal is to obtain an adapted policy for the target $G$ that ensures high utility $f_{\boldsymbol{\omega}_g}(\boldsymbol{r})$ and meets the constraints with safety threshold $\boldsymbol{\beta}_g$ by leveraging demonstration set $\mathcal{B}_G$ during the deployment phase.

## 4.2 Offline Adaptation for the Unconstrained Case

First, we set aside the constraints in CMO-MDP and focus on the unconstrained version. Our proposed framework, Preference Distribution Offline Adaptation (PDOA), solves the offline adaptation problem under unconstrained settings in two steps: **1)** learning a set of policies that respond to various preferences during training; and then **2)** adapting a distribution of target preferences based on given demonstrations during deployment.

In the first part, we directly apply existing offline MORL algorithms on the dataset $\mathcal{D}$ to obtain a set of policies $\hat{\pi}_{\boldsymbol{\omega}}$ that respond to varying preferences $\boldsymbol{\omega}$. In the second phase, we propose to model the distribution of target preferences and then utilize this distribution to obtain a reliable estimation of target preference. Specifically, we consider the posterior probability of the target preference $\boldsymbol{\omega}_g$ with regard to demonstration set $\mathcal{B}_{\boldsymbol{\omega}_g}$, i.e., $P(\boldsymbol{\omega}_g|\mathcal{B}_{\boldsymbol{\omega}_g}, \mathcal{D}) = \frac{P(\mathcal{B}_{\boldsymbol{\omega}_g}|\boldsymbol{\omega}_g, \mathcal{D})P(\boldsymbol{\omega}_g|\mathcal{D})}{P(\mathcal{B}_{\boldsymbol{\omega}_g}|\mathcal{D})}$. Here $P(\mathcal{B}_{\boldsymbol{\omega}_g}|\boldsymbol{\omega}_g, \mathcal{D})$ represents the probability that the optimal $\pi_{\boldsymbol{\omega}_g}^*$ generates samples $\mathcal{B}_{\boldsymbol{\omega}_g}$ in the real environment $\mathcal{P}(s', \boldsymbol{r}|s, a)$. Due to the inaccessibility of $\mathcal{P}$ and $\pi_{\boldsymbol{\omega}}^*$ under offline settings, we replace them with the empirical dynamics $\hat{\mathcal{P}}_{\boldsymbol{\omega}}(s'|s, a)$ and its corresponding optimal policy $\hat{\pi}_{\boldsymbol{\omega}}^*$, which can be obtained using offline data $\mathcal{D}$ during training. Therefore, the preference posterior distribution can be approximated by

$$P(\boldsymbol{\omega}|\mathcal{B}_{\boldsymbol{\omega}_g}, \mathcal{D}) = \frac{P(\mathcal{B}_{\boldsymbol{\omega}_g}|\boldsymbol{\omega}, \mathcal{D})P(\boldsymbol{\omega}|\mathcal{D})}{P(\mathcal{B}_{\boldsymbol{\omega}_g}|\mathcal{D})} \approx \frac{P(\mathcal{B}_{\boldsymbol{\omega}_g}|\hat{\mathcal{M}}_{\boldsymbol{\omega}}, \hat{\pi}_{\boldsymbol{\omega}}^*, \mathcal{D})P(\boldsymbol{\omega}|\mathcal{D})}{P(\mathcal{B}_{\boldsymbol{\omega}_g}|\mathcal{D})}$$
$$\propto P(\boldsymbol{\omega}|\mathcal{D}) \prod_{i=1}^{M} P_{\hat{\pi}_{\boldsymbol{\omega}}^*}(s_i) \hat{\pi}_{\boldsymbol{\omega}}^*(a_i|s_i) \hat{\mathcal{P}}_{\boldsymbol{\omega}}(s_i', \boldsymbol{r}_i|s_i, a_i), \tag{5}$$

where $(s_i, a_i, r_i, s_i') \in \mathcal{B}_{\boldsymbol{\omega}_g}$. One challenge in Eq. (5) is the requirement of explicitly modeling the state distribution $P_{\hat{\pi}_{\boldsymbol{\omega}}^*}(s_i)$ and the transition probability $\hat{\mathcal{P}}_{\boldsymbol{\omega}}(s_i', \boldsymbol{r}_i|s_i, a_i)$. Another concern is that demonstrations $\mathcal{B}_{\boldsymbol{\omega}_g}$ with the target preference $\boldsymbol{\omega}_g$ can be out-of-distribution samples for the estimation of $\hat{\pi}_{\boldsymbol{\omega}}^*$ and $\hat{\mathcal{M}}_{\boldsymbol{\omega}}$, leading to considerable discrepancy between estimated $P_{\hat{\pi}_{\boldsymbol{\omega}}^*}(s_i)$, $\hat{\mathcal{P}}_{\boldsymbol{\omega}}(s_i', \boldsymbol{r}_i|s_i, a_i)$ and their real-world counterparts. This challenge is pronounced in multi-objective settings due to significant differences in the trajectory distributions of the optimal policy under various preferences. Therefore, following previous offline adaptation works [Ghosh et al., 2022, Hong et al.,

2022], we opt to approximate $\log P_{\hat{\pi}_{\boldsymbol{\omega}}^*}(s_i)\hat{\mathcal{P}}_{\boldsymbol{\omega}}(s_i', \boldsymbol{r}_i|s_i, a_i)$ with a surrogate defined by TD error of value models of $\hat{\pi}_{\boldsymbol{\omega}}^*$ in $\hat{\mathcal{M}}_{\boldsymbol{\omega}}$:

$$r_{\boldsymbol{\omega}}^{\mathrm{TD}}(s, a, \boldsymbol{r}, s') = -\delta\|\boldsymbol{Q}(s, a, \boldsymbol{\omega}) - (\boldsymbol{r} + \gamma\boldsymbol{V}(s', \boldsymbol{\omega}))\|_2^2, \tag{6}$$

where $\delta$ is a hyperparameter. This approximation makes sense because $r_{\boldsymbol{\omega}}^{\mathrm{TD}}(s, a, \boldsymbol{r}, s')$ not only measures how likely the sample $(s, a, \boldsymbol{r}, s')$ occurred during training for preference $\boldsymbol{\omega}$ but also aligns with the estimated dynamics $\hat{\mathcal{P}}_{\boldsymbol{\omega}}$. In other words, $r_{\boldsymbol{\omega}}^{\mathrm{TD}}(s, a, \boldsymbol{r}, s')$ has a high value if $(s, a, \boldsymbol{r}, s')$ is an in-distribution sample relative to the training datasets under preference and occurs with a high probability in $\hat{\mathcal{M}}_{\boldsymbol{\omega}}$.

Then, we fit the posterior $P(\boldsymbol{\omega}|\mathcal{B}_{\boldsymbol{\omega}_g}, \mathcal{D})$ with a Gaussian distribution $\mathcal{N}(\boldsymbol{\mu}, \boldsymbol{\sigma}\boldsymbol{I})$ with parameters $\boldsymbol{\mu} \in R^K$ and $\boldsymbol{\sigma} \in R_+^K$ by minimizing their Kullback-Leibler divergence, leading to the following adaptation loss:

$$\begin{aligned}
\mathcal{L}_{\mathrm{adpt}}(\boldsymbol{\mu}, \boldsymbol{\sigma}) =& D_{\mathrm{KL}}(p_\theta(\boldsymbol{\omega})\|P(\boldsymbol{\omega}|\mathcal{B}_{\boldsymbol{\omega}_g}, \mathcal{D}))/M + \eta(\|\boldsymbol{\mu}\|_1 - 1)^2 \\
=& -\mathbb{E}_{\boldsymbol{\omega}\sim\mathcal{N}(\boldsymbol{\mu}, \boldsymbol{\sigma}\boldsymbol{I})}\left[\frac{1}{M}\sum_{i=1}^M\left[r_{\boldsymbol{\omega}}^{\mathrm{TD}}(s_i, a_i, \boldsymbol{r}_i, s_i') + \log\hat{\pi}_{\boldsymbol{\omega}}^*(a_t|s_t, \boldsymbol{\omega})\right] + \\
& \frac{\log P(\boldsymbol{\omega}|\mathcal{D})}{M}\right] - \frac{H(p_\theta)}{M} + \eta(\|\boldsymbol{\mu}\|_1 - 1)^2,
\end{aligned} \tag{7}$$

where the term $\eta(\|\boldsymbol{\mu}\|_1 - 1)^2$ with hyperparameter $\eta$ regularizes $\|\boldsymbol{\mu}\|_1$ to be close to 1. We approximate the prior distribution $P(\boldsymbol{\omega}|\mathcal{D})$ with a Gaussian distribution fit to the behavioral preferences of training data $\mathcal{D}$. Eq. (7) indicates that $\boldsymbol{\omega}$ is likely to be the target preference $\boldsymbol{\omega}_g$ if it corresponds to high $r_{\boldsymbol{\omega}}^{\mathrm{TD}}(s_i, a_i, \boldsymbol{r}_i, s_i')$, $\log\hat{\pi}_{\boldsymbol{\omega}}^*(a_t|s_t, \boldsymbol{\omega})$ and $P(\boldsymbol{\omega}|\mathcal{D})$, which means: 1) the demonstration set $\mathcal{B}_{\boldsymbol{\omega}_g}$ appears with high probability in both the training set and $\hat{\mathcal{M}}_{\boldsymbol{\omega}}$; 2) $\hat{\pi}_{\boldsymbol{\omega}}^*(a_t|s_t, \boldsymbol{\omega})$ is likely to generate the actions in $\mathcal{B}_{\boldsymbol{\omega}_g}$; and 3) $\boldsymbol{\omega}$ stays within the prior preference distribution with a high probability. We justify the effectiveness of this approach with the empirical results in Appendix A.9, where $r_{\boldsymbol{\omega}}^{\mathrm{TD}}(s_i, a_i, \boldsymbol{r}_i, s_i')$ (called TD reward) and $\hat{\pi}_{\boldsymbol{\omega}}^*(a_t|s_t, \boldsymbol{\omega})$ (called action likelihood reward) shows a strong correlation with the real target preference.

**Implementation** The first part of our framework PDOA is instantiated with state-of-the-art MORL algorithms: MODF and the best algorithm MORvS in PEDA mentioned in Section 3.2, to learn a set of policies that responds to various preferences. We refer to these two instances as **PDOA [MODF]** and **PDOA [MORvS]**. The well-trained policies and their corresponding value functions obtained by MORL algorithms during training are utilized to calculate the action likelihood reward $\hat{\pi}_{\boldsymbol{\omega}}^*(\cdot)$ and TD reward $r_{\boldsymbol{\omega}}^{\mathrm{TD}}(\cdot)$ in Eq. (7). During deployment, the preference distribution $p_\theta(\boldsymbol{\omega})$ is updated through the adaptation loss (7) on the demonstration set $\mathcal{B}_{\boldsymbol{\omega}_g}$. Finally, we obtain an adapted target preference $\boldsymbol{\omega}_a = \boldsymbol{\mu}/\|\boldsymbol{\mu}\|_1$ and policy $\hat{\pi}_{\boldsymbol{\omega}}^*(\cdot|\boldsymbol{\omega}_a)$ that aligns with the real target preference $\boldsymbol{\omega}_g$ implied in demonstrations $\mathcal{B}_{\boldsymbol{\omega}_g}$. More details about implementation can be found in Appendix A.5.

### 4.3 Extension to Constrained Settings

Then, we consider a natural extension of our MORL adaptation framework to constrained settings. Under the linear preference setting, the constrained MORL problem in Eq. (1) can be converted to its dual form with zero duality gap [Paternain et al., 2019]:

$$\min_{\boldsymbol{\lambda}\in R_+^K}\max_\pi\mathbb{E}_\pi\Big[\sum_t\boldsymbol{\omega}_g^\mathsf{T}\boldsymbol{r}_t - \boldsymbol{\lambda}^\mathsf{T}(\sum_t\boldsymbol{c}_t - \boldsymbol{\beta}_g)\Big]. \tag{8}$$

Denoting $\lambda^*$ as the solution of this problem, the dual problem 8 can be rewritten as:

$$\max_\pi\mathbb{E}_\pi\Big[\sum_t[\boldsymbol{\omega}_g^\mathsf{T}, \boldsymbol{\lambda}^{*\mathsf{T}}]\cdot[\boldsymbol{r}_t^\mathsf{T}, -\boldsymbol{c}_t^\mathsf{T}]^\mathsf{T}\Big]. \tag{9}$$

This formulation means that the constrained MORL problem under safety threshold $\boldsymbol{\beta}$ is uniquely equivalent to an unconstrained problem of finding the optimal policy under an extended preference $\tilde{\boldsymbol{\omega}}_g = [\boldsymbol{\omega}_g^\mathsf{T}, \boldsymbol{\lambda}^{*\mathsf{T}}]^\mathsf{T}/\|[\boldsymbol{\omega}_g^\mathsf{T}, \boldsymbol{\lambda}^{*\mathsf{T}}]^\mathsf{T}\|_1$ among extended $N + K$ objectives $\tilde{\boldsymbol{r}} = [\boldsymbol{r}_t^\mathsf{T}, -\boldsymbol{c}_t^\mathsf{T}]^\mathsf{T}$. However, solving for the extended preference is challenging and requires the real safety thresholds, which

are inaccessible in our setting. Nevertheless, Section 4.2 provides an approach to infer the target preferences from the demonstration set, helping us circumvent this challenge. Therefore, we can convert a constrained MORL problem to an unconstrained MORL problem, where the vector reward is defined as $\tilde{r} = [r_t, -c_t]$ and dataset $\mathcal{D}$ is augmented to $\hat{\mathcal{D}}$. Here, $\tilde{r}_{1:N} = r$ corresponds to $N$ unconstrained objectives, while $\tilde{r}_{N+1:N+K} = -c$ associated with $K$ constrained objectives. One issue with this approach is how to set the behavior preference for augmented dataset $\hat{\mathcal{D}}$. We present an effective scheme for approximating behavioral preferences in Appendix A.5.

However, one concern about this approach is the estimation error between the adapted preference $\tilde{\omega}_a$ obtained through Eq. (7) and the real preference $\tilde{\omega}_g$ associated with the problem 9 due to the insufficiency of demonstrations. This discrepancy can lead to constraint violations when the preference weight of the adapted preference $\tilde{\omega}_a$ on a constrained objective is less than the one of $\tilde{\omega}_g$ (i.e., $[\tilde{\omega}_a]_i < [\tilde{\omega}_g]_i$ for any $i \in \{N+1, ..., N+K\}$). Therefore, we propose to apply a conservative estimate of preference weights on constrained objectives by neglecting the minimum value with probability $1 - \alpha$ in the adaptation distribution $\tilde{\omega} \sim \mathcal{N}(\mu, \sigma)$. Specifically, the preference weights on the $i^{\text{th}}$ objective is estimated as follows:

$$b_i = \text{CVaR}_\alpha([\tilde{\omega}]_i) = \frac{1}{\alpha} \int_{1-\alpha}^1 \text{VaR}_u([\tilde{\omega}]_i)\mathrm{d}u = \mu_i + \sigma_i \frac{\varphi(\Phi^{-1}(1-\alpha))}{\alpha}, N < i \leq N+K, \quad (10)$$

and $b_i = \mu_i$ for $1 \leq i \leq N$. Here, VaR is the value of risk and CVaR is the conditional one, and $\varphi(x) = \frac{1}{\sqrt{2\pi}} \exp(-\frac{x^2}{2})$ is the standard normal p.d.f., and $\Phi(x)$ is the standard normal c.d.f.. The parameter $\alpha$ in Eq. (10) controls the conservatism of the estimation of preference weights. For all constrained objectives, $\alpha < 1$ results in a conservative estimate $b_i > \mu_i$, reducing the risk of constraint violation. Additionally, as the number of demonstrations increases, the standard deviation $\sigma_i$ decreases, and thus $b_i$ will move towards $\mu_i$, resulting in less conservatism. In the end, we obtain the adapted target preference $\tilde{\omega}_a = b/\|b\|_1$ and the adapted policy $\pi(s|a, \tilde{\omega}_a)$.

## 5 Experiment

In this section, we conduct several experiments on classical MORL environments in Section 5.1 and safe RL tasks in Section 5.2 to evaluate our framework in achieving preference alignment and approaching constraint satisfaction, respectively. Then, we test our method on a set of new constrained MORL (denoted as CMORL) tasks in Section 5.3. Finally, we discuss the effectiveness of the components of our method in Section 5.4.

**Environments and Datasets**    In Section 5.1, we utilize the D4MORL datasets [Zhu et al., 2023] for MORL experiments, which involve two conflict objectives and a variety of behaviors based on preferences in multi-objective MuJoCo tasks. In Section 5.2, all algorithms are trained on the datasets from DSRL benchmark [Liu et al., 2023b], which involve a constrained objective and an unconstrained objective in BulletSafeGym tasks [Gronauer, 2022]. In Section 5.3, we develop a set of CMORL tasks by incorporating an additional velocity constraint to the original multi-objective MuJoCo environments and collect datasets from these tasks. Thus, these environments contain two unconstrained objectives and one constrained objective. More details about these environments and datasets can be found in Appendix A.2. We construct training sets and demonstration sets based on the above datasets and present the details on Appendix A.3.

**Evaluation Protocols**    For evaluation, we define a target set $\mathcal{T} = \{G\}$, with each target $G$ corresponding to a demonstration set $\mathcal{B}_G$ that meets the target $G$. For MORL tasks in Section 5.1, $G$ is a target preference $\omega_g$. All target preferences are selected from the dataset's preference support set at a fixed interval 0.01 (i.e., $\omega_g = [0.01k, 1 - 0.01k]$, s.t. $\min \hat{\omega} \preceq \omega_g \preceq \max \hat{\omega}$ where $k = 0, .., 100$ and $\hat{\omega}$ is the behavioral preference of the dataset). For safe RL tasks in Section 5.2, $G$ is a safety threshold $\beta_g$. All safety thresholds are set to 6 equidistant points within the range of possible thresholds. The target of CMORL tasks in Section 5.3 is $(\omega_g, \beta_g)$, the Cartesian product of the target preference spaced at 0.1 interval and 6 equidistant safety thresholds.

We utilize $\mathcal{B}_G$ to generate an adapted policy for each target $G$ and gather 5 trajectories from environments to assess its expected vector return. For tasks without constraints, we present the average utility and Hypervolume metrics to demonstrate the overall performance across all targets and the diversity

of adapted policies. For tasks with constraints, we group the adapted policies by safety thresholds and report the maximum cost return on constrained objectives and average utility and Hypervolume on unconstrained objectives for each group. We perform 3 runs with various seeds and report average performance along with 1-sigma error bar. More details about metrics can be found in Appendix A.4.

**Comparative Algorithms**   The algorithms for comparison are divided into two categories. The first category is preference/threshold-agnostic baselines that have no access to the real target preferences or safety thresholds implied in the demonstrations, including: 1) **BC-Finetune**, where the policy is learned on training dataset via behavior cloning and is fine-tuned on demonstration sets and 2) **Prompt-MODT**, a multi-objective version of Prompt-DT [Xu et al., 2022a] that transforms a multi-objective scenario into a multi-task problem via preference-based division and achieves offline adaptation for various tasks by taking demonstrations as prompts of the transformer. The second category is preference/threshold-informed baselines that have access to the implied preferences and thresholds and thus serve as the oracle benchmark, including: 1) original **MODF** and **MORvS** for MORL tasks, 2) **CDT** [Liu et al., 2023a] for safe RL tasks, which makes decisions based on given safety thresholds to ensure constraint satisfaction for various thresholds. For CMORL tasks, instead of adapting the preference through demonstrations, we enumerate all possible augmented preferences of MODF with a small interval and report the best performance under these preferences as the oracle. More details about comparative algorithms are placed in Appendix A.4.

### 5.1   Preference Alignment for MORL Tasks

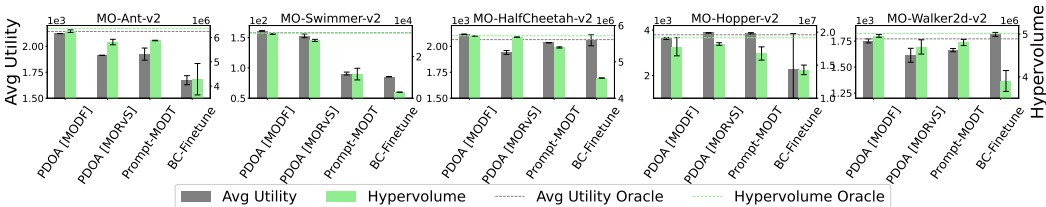

Figure 1: Results on D4MORL Amateur datasets. Higher average utility and Hypervolume are preferable. The dashed lines represent the best performance between the original MODF and MORvS.

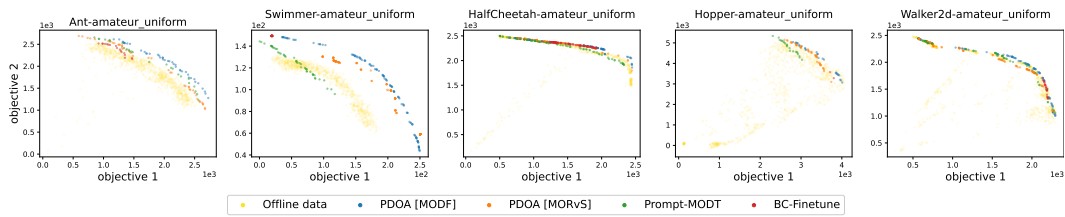

Figure 2: Pareto fronts of different algorithms on D4MORL Amateur datasets. Each point represents an adapted policy for a specific unknown target preference.

The average utility and Hypervolume of all algorithms are shown in Figure 1. Our method demonstrates superior overall performance and matches the oracle performance of preference-informed methods. Moreover, we present the Pareto fronts of different algorithms in Figure 2. Our method produces a broader and expanding Pareto front compared to BC-Finetune and Prompt-MODT, which indicates that the adapted policies of our method exhibit higher diversity and better performance. Meanwhile, we observe that the adapted policies obtained by BC-Finetune cluster around the behavior-cloned policy, and thus BC-Finetune obtains the narrow Patero fronts and low Hypervolume, which indicates the difficulty of changing the policy's preference through simple fine-tune with limited samples. Besides, Prompt-MODT also yields a restricted Pareto front, which can be attributed to the difficulty of partitioning tasks based on preferences. Fine-grained divisions result in data insufficiency for each task, while coarse-grained divisions lead to multiple preference behaviors being grouped into

a single task, both of which can hurt the policy performance and diversity. In contrast to BC-Finetune and Prompt-MODT, our method explicitly learns a set of policies with various preferences through MORL, thereby ensuring policy diversity. Once the target preferences are accurately identified, we can generate policies with diverse behaviors to meet various target preferences. To verify the capability of our method in preference alignment, we show the differences between the adapted preferences obtained by our method and the real target preferences in Figure 3, where we can observe a strong consistency between the adapted preferences and the real ones, especially on PDOA [MODF]. We additionally present the performance, Pareto fronts and preference comparison on the D4MORL expert datasets in Appendix A.6.1, which are consistent with the results in this section.

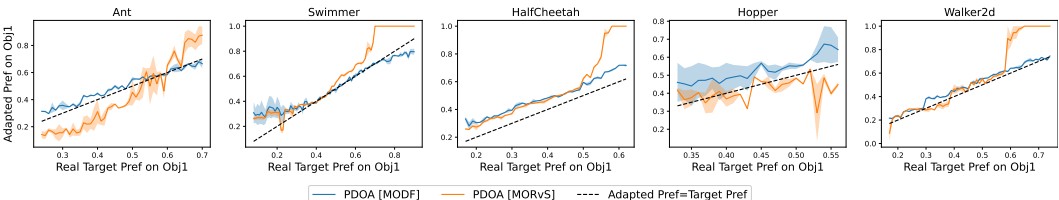

Figure 3: The comparison between the real target preferences and the adapted preferences.

## 5.2 Constraint Satisfaction for Safe RL Tasks

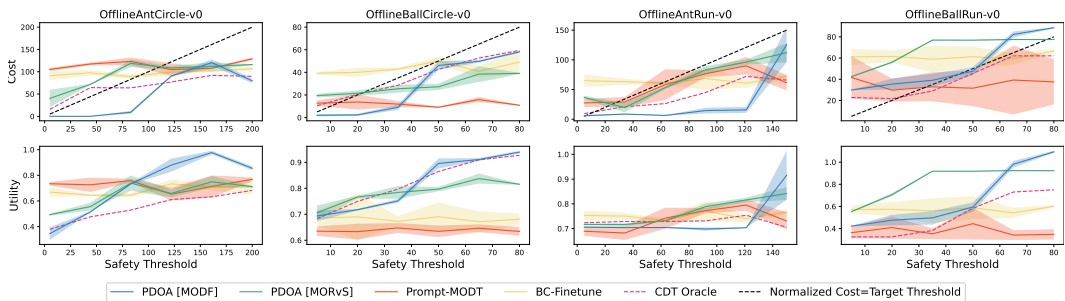

Figure 4: The adapted policies' cost and utility of each algorithm under various safety thresholds. Here, the utility is the normalized reward, since there is only one unconstrained objective in DSRL tasks. The points above the black dashed line represent the policies that violate the constraints.

Figure 11 in Appendix A.7 presents the Pareto fronts of MODF and MORvS, i.e., the expected costs and expected rewards under various preferences, for safe RL tasks, which demonstrates that MORL algorithms can learn a variety of policies that meet various safety thresholds. Then, the reward and cost of all algorithms are shown in Figure 4. We also present the performance and Pareto fronts on other tasks in Appendix A.7. These results show that, even though the safety thresholds are inaccessible, PDOA [MODF] achieves relatively safe performance under various safety thresholds compared to other baselines, closely matching the oracle performance of the threshold-informed baseline CDT. Even with very tight safety thresholds, PDOA [MODF] can achieve constraint satisfaction or experience few constraint violations. Meanwhile, its reward performance is comparable to or even exceeds that of CDT. However, PDOA [MORvS] performs poorly on the DSRL datasets because MORvS requires accurate predictions of the target return for each preference, which is challenging due to the abundance of suboptimal trajectories in the DSRL datasets. Additionally, BC-Finetune and Prompt-MODT, which align behaviors without considering the constraints on constrained objectives, exhibit insufficient policy diversity and constraint violations in most environments.

## 5.3 Evaluation for Constrained MORL Tasks

The results on CMO datasets are shown in Figure 5. The CMO tasks involve more objectives than previous MORL and safe RL tasks, posing a challenge in identifying and aligning behaviors.

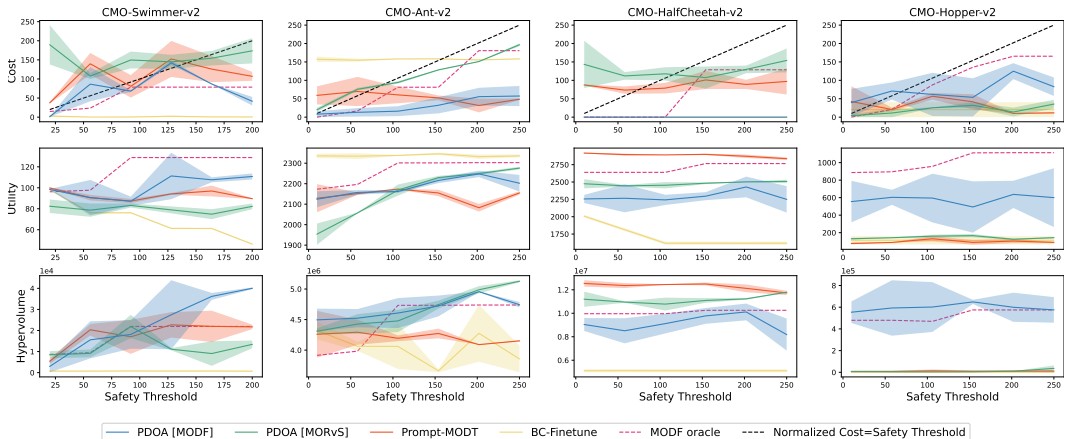

Figure 5: The maximum cost, the average utility and Hypervolume over all targets with a specific safety threshold.

This is because behaviors with various preferences increase exponentially as the number of objectives increases. We can observe that in most environments, the performance of BC-Finetune and Prompt-MODT shows low Hypervolume and a weak correlation with the safety threshold, indicating that they cannot align with the desired behaviors and exhibit high policy diversity. Nevertheless, PDOA [MODF] approaches the oracle performance with high average utility, Hypervolume, and few constraint violations, which is consistent with the results presented in Sections 5.1 and 5.2.

## 5.4 Ablation Study

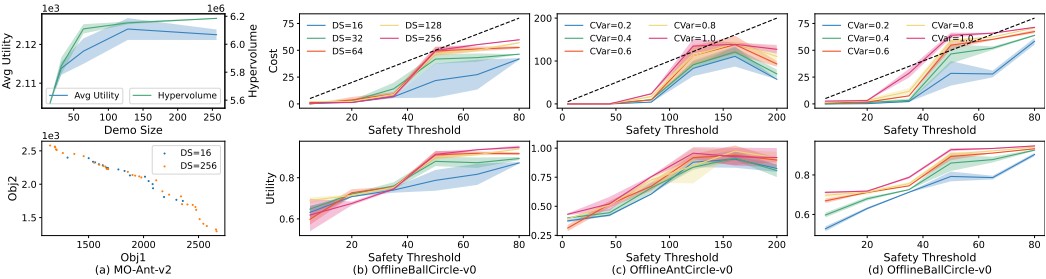

Figure 6: The performance across different demonstration sizes (abbr. DS) in Figure (a)(b) and the performance under various conservatism parameters (abbr. CVaR) in Figure (c)(d).

In this section, we aim to figure out: 1) the impact of the quantity of demonstrations on adaptation performance, and 2) the influence of the conservatism parameter $\alpha$ of Eq. (10) on safety performance. We conduct a set of ablation experiments for PDOA [MODF] on several MORL and safe RL tasks and present the results in Figure 6. In Figure 6 (a)(b), as the demonstration size (DS) increases, our method achieves better average utility, Hypervolume for MORL tasks and higher reward for safe RL tasks. Nevertheless, even with a very small number of demonstrations (DS = 16), the adapted policies generated by our method still exhibit sufficient diversity and safety. Figure 6 (c)(d) demonstrate that by increasing the conservatism parameters, we can further reduce constraint violations and ultimately achieve constraint satisfaction, which validates the effectiveness of our conservatism mechanism in enhancing safety.

# 6 Conclusion

In this paper, we present an offline adaptation framework Preference Distribution Offline Adaptation (PDOA) for constrained MORL problems where we assume no access to real target preferences or safety thresholds and only a few demonstrations with expected behaviors are available in our framework. For unconstrained MORL scenarios, we propose to 1) employ MORL methods to train a set of preference-varying policies, and then 2) align the preferences of adapted policies with expected behaviors. Furthermore, we expand our framework to accommodate constraints on specific objectives by transforming constrained problems into unconstrained counterparts. Additionally, we introduce a conservatism mechanism for preference estimation on constrained objectives to mitigate potential constraint violations. Empirical results on MORL and safe RL tasks illustrate the capability of our framework in generating diverse policies aligning with expected behaviors and approach constraint satisfaction through conservative preference estimation.

**Limitation** The PDOA framework involves multiple steps of preference sampling and gradient updates, which can lead to additional computational burden and latency during deployment. Besides, although the manual design of preferences and safety thresholds is not necessary, the process of constructing demonstrations may still involve human expertise.

## Acknowledgements

We gratefully acknowledge the support from the National Natural Science Foundation of China (No. 62076259, 62402252), the Fundamental and Applicational Research Funds of Guangdong Province (No. 2023A1515012946), the Fundamental Research Funds for the Central Universities Sun Yat-sen University, and the Pengcheng Laboratory Project (PCL2023A08, PCL2024Y02).

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

# A Appendix / supplemental material

## A.1 Connections and Differences to Other Fields

**Multi-task RL**   While the constrained MORL setting we focus on can align with the multi-task framework by treating policy learning under various preferences or thresholds as distinct tasks, several key differences distinguish our study from previous multi-task RL settings: (1) One primary motivation of our work is to circumvent the handcrafted design of target preferences, whereas previous studies [Mitchell et al., 2021, Xu et al., 2022a] on multi-task RL mainly focus on policy generalization to new tasks. (2) We achieve offline adaptation by updating the target preference distribution rather than the policies themselves, which distinguishes our framework from offline meta-RL methods. This adaptation pipeline not only avoids potential performance degradation due to policy parameter shifts but also allows us to incorporate conservatism to mitigate constraint violations. (3) Accurately identifying various preferences and constraints implied in demonstrations during adaptation is challenging as these demonstrations could come from the same dynamics and policies with similar preferences.

**Multi-constraint RL**   Existing research on multi-constraint RL typically explores the correlation between multiple constraints from gradient perspectives [Yao et al., 2024b, Kim et al., 2024]. Some works [Guan et al., 2024b] attempt to meet different types of constraints simultaneously. In contrast, our approach, from the policy diversity perspective, aims to find appropriate policies for various safety requirements and thus places less focus on constraint correlation for a specific threshold. Nevertheless, previous studies on multiple constraint RL will provide important insights into managing potential correlations between multiple objectives during preference-conditioned policy learning, thereby improving overall performance.

**Inverse RL**   Existing IRL methods [Arora and Doshi, 2021] typically assume no knowledge about the reward function and focus on learning reward functions from mere demonstrations through apprenticeship learning, maximum entropy optimization or adversarial learning. Some of these methods have been applied in MORL [Takayama and Arai, 2022] or safe RL [Malik et al., 2021]. In contrast to these studies, our work assumes knowledge of the reward information on different objectives and pays more attention to identifying the preferences of demonstrations during deployment than learning a specific decisive reward during training.

**Bayesian RL**   Bayesian RL [Ghavamzadeh et al., 2015] is widely used for promoting online exploration and dealing with uncertainty in model-based RL or offline RL, while it serves as a cornerstone of our theoretical derivation, allowing us to incorporate prior preference information from datasets and infer the expected preferences by estimating the preference posterior based on a few demonstrations. Ghosh's work [Ghosh et al., 2022] is the most related Bayesian RL study that inspires our approach. Nevertheless, it aims to reduce the dynamics uncertainty caused by limited offline samples, while our approach focuses on utilizing Bayesian inference to solve the uncertainty of the target preference distribution based on a few demonstrations.

**Offline Imitation Learning**   Each adaptation can be viewed as the imitation of a expert policy with a certain preference. Therefore, offline Imitation Learning (IL) is a potential approach to solving the problem by applying an off-the-shelf offline IL algorithm [Xu et al., 2022b, Kim et al., 2022], which enables the combination of sub-optimal trajectories in the offline dataset with expert demonstrations provided by users to imitate expert behaviors. However, applying offline IL to our problem presents several challenges. We conduct comparative experiments in Appendix A.8 to further compare algorithms and illustrate these challenges.

## A.2 Environments and Datasets Details

**Environments and Datasets for Multi-objective RL**   We utilize the D4MORL dataset [Zhu et al., 2023] collected from multi-objective MuJoCo environments by a set of preference-varying policies [Xu et al., 2020]. We consider 5 MuJoCo tasks including HalfCheetah, Ant, Hopper, Walker2d, and Swimmer. Among them, the HalfCheetah, Walker2d, and Swimmer tasks involve

conflicting objectives of running speed and energy saving. The Ant task includes two speed objectives in both the x and y axes, while the Hopper considers running and jumping objectives. For each task, we test algorithms on two types of datasets: Expert and Amateur, which are collected by pure expert policies and noise-injected expert policies, respectively.

**Environments and Datasets for Safe RL**  We utilize the datasets in DSRL benchmark [Liu et al., 2023b] that are collected by a set of behavior policies trained under various safe thresholds. We select 8 BulletSafetyGym tasks [Gronauer, 2022] for safe RL experiments, which involve 4 agents (Ball, Car, Drone, Ant) and 2 task goals (Run, Circle). In these tasks, the goal of the reward function is to navigate to target positions or complete circuits as quickly as possible, while the costs are incurred when the agent enters risky areas.

**Environments and Datasets for Constrained Multi-objective RL**  We develop a set of CMORL tasks, namely CMO-Hopper, CMO-HalfCheetah, CMO-Ant and CMO-Swimmer. These tasks are constructed based on the multi-objective MuJoCo environments [Xu et al., 2020] and augmented by an additional constraint on robotics' velocity, which is consistent with the constraint setting of Safety Gymnasium tasks in the DSRL benchmark. When the agent's moving velocity exceeds a specific value, it will receive 1 cost. To gather datasets for these tasks, we create a behavioral preference set and utilize TRPO+Lagrangian [Liu et al., 2023b] to train individual policies with the goal of maximizing utility under each preference in this set while accounting for a specified velocity constraint. The safety threshold of the velocity constraint is adjusted throughout the training process to encompass a range of potential safety thresholds. The details for each CMO task are shown in Table 1. The replay buffers collected during the training of these policies constitute the CMO datasets. We present Pareto fronts of CMO datasets under different cost return intervals in Figure 7, illustrating that CMO datasets encompass a wide range of behaviors catering for various preferences and safety thresholds.

Table 1: Details of each CMO task.

| Details | CMO-Ant | CMO-Hopper | CMO-HalfCheetah | CMO-Swimmer |
|---|---|---|---|---|
| behavioral preference Set | $[0.5, 0.5], [0.6, 0.4], [0.7, 0.3], [0.8, 0.2], [0.9, 0.1], [1.0, 0.0]$ | | | |
| Cost function | Moving Velocity (cost=1 if the velocity exceeds a specific value) | | | |
| Threshold Range | $[10, 250]$ | | | $[10, 200]$ |
| Objectives | Speed on X-axis and Y-axis | Moving speed and jumping height | Moving speed and energy consumption | |

### A.3  Construction of Training Set and Demonstration Set

(1) For MORL experiments in Section 5.1, each target preference $\omega_g$ is associated with a demonstration set $\mathcal{B}_{\omega_g}$, which contains $M$ transitions[2] randomly sampled from $K$ trajectories in D4MORL Expert datasets with behavioral preferences closest to the target preference $\omega_g$. (2) For safe RL experiments in Section 5.2, each target safety threshold $\beta_g$ is associated with a demonstration set $\mathcal{B}_{\beta_g}$, which contains $M$ transitions randomly sampled from $K$ trajectories in DSRL datasets with the highest utility among all safe trajectories (i.e., these trajectories' cumulative cost return is less than $\beta$). (3) For CMORL experiments in Section 5.3, each combination of target preference $\omega_g$ and safety threshold $\beta_g$ is associated with a demonstration set $\mathcal{B}_{\omega_g, \beta_g}$, which consists of $M$ transitions randomly sampled from $K$ trajectories in CMO datasets that (i) have highest utility among all safe trajectories and (ii) originate from the replay buffer of the behavior policy with preference $\omega_g$. The unselected trajectories in datasets constitute the training set. In our experiments, $M = 128$ and $K = 2$ by default.

---

[2]For Prompt-MODT, which required trajectory segments as prompts during adaptation, we sample a trajectory segment with length=$M$ to constitute the demonstration set.

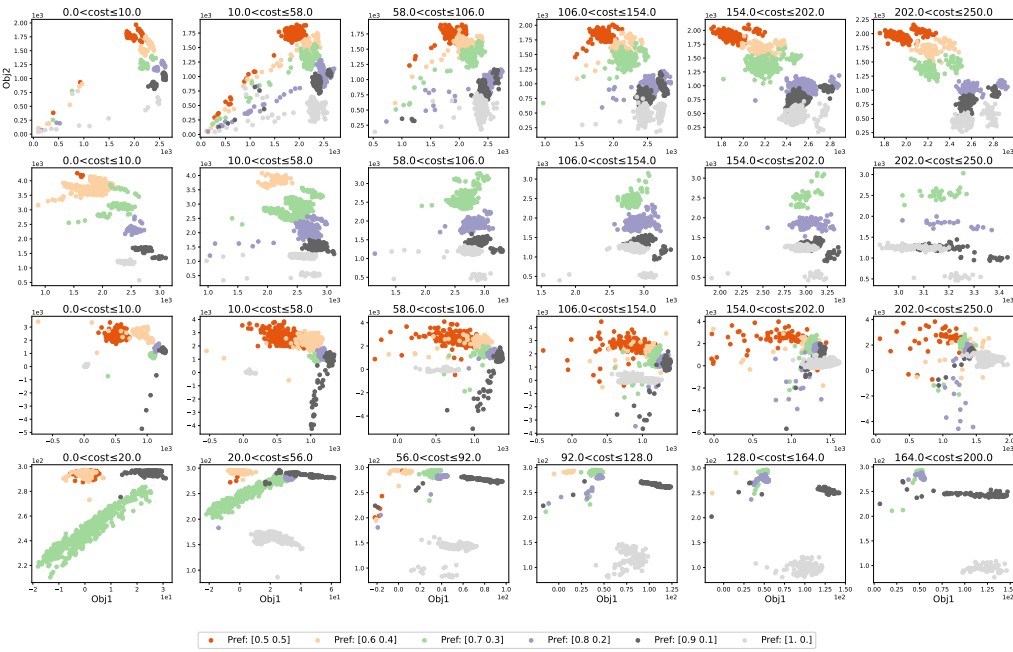

Figure 7: The reward vector distribution of CMO datasets under various safety thresholds.

### A.4 Evaluation Metrics

We denote $R(\pi_G)$ as the estimated return vector of the policy $\pi_G$ adapted for target $G$ in the target set $\mathcal{T}$.

**Average Utility**   For unconstrained scenarios, $G = \boldsymbol{\omega}_g$ and average utility is calculated by:

$$U = \frac{1}{|\mathcal{T}|} \sum_{\boldsymbol{\omega}_g \in \mathcal{T}} \boldsymbol{\omega}_g^{\mathsf{T}} R(\pi_{\boldsymbol{\omega}_g}). \tag{11}$$

For constrained scenarios, $G = (\boldsymbol{\omega}_g, \boldsymbol{\beta})$. We denote $\mathcal{T}_{\boldsymbol{\beta}_g} = \{\boldsymbol{\omega}_g | (\boldsymbol{\omega}_g, \boldsymbol{\beta}_g) \in \mathcal{T}\}$ as subset of $\mathcal{T}$ that groups all target preference $\boldsymbol{\omega}_g$ by safety threshold $\boldsymbol{\beta}_g$. The average utility is a function of $\boldsymbol{\beta}_g$:

$$U(\boldsymbol{\beta}_g) = \frac{1}{|\mathcal{T}_{\boldsymbol{\beta}_g}|} \sum_{\boldsymbol{\omega}_g \in \mathcal{T}_{\boldsymbol{\beta}_g}} \boldsymbol{\omega}_g^{\mathsf{T}} R(\pi_{\boldsymbol{\omega}_g, \boldsymbol{\beta}_g}). \tag{12}$$

This metric reflects the overall performance of the adapted policies across all target preferences.

**Hypervolume**   We first consider the unconstrained settings and introduce the adapted Pareto set $P$ that contains all adapted policies that are not dominated by any other adapted policy. Hypervolume measures the volume enclosed by the returns of policies in the adapted Pareto set $P$:

$$\text{HV} = \int_{R^n} \mathbb{1}_{H(P, r_0)}(\boldsymbol{z}) \mathrm{d}\boldsymbol{z}, \tag{13}$$

where $H(P, \boldsymbol{r}_0) = \{\boldsymbol{z} \in R^n | \exists \pi \in P : \boldsymbol{r}_0 \preceq \boldsymbol{z} \preceq R(\pi)\}$ and $\mathbb{1}_{H(P, r_0)}$ equals 1 if $\boldsymbol{z} \in H(P)$ and 0 otherwise. For constrained scenarios, we group all target preferences by safety threshold and report Hypervolume on each group. The Hypervolume reflects not only the overall performance across all target preferences but also the diversity of the policies, as a high diversity of policies corresponds to a broad Pareto front and consequently a high Hypervolume.

**Sparsity**, which measures the density of policies in the approximated Pareto set, is also commonly used for evaluating MORL performance. A lower sparsity indicates a denser Pareto front, which

---

**Algorithm 1** Preference Distribution Offline Adaptation

---

 1: ### Training Phase
 2: **Input:** Offline dataset $\mathcal{D}$
 3: Train policy-conditioned policy $\hat{\pi}_{\boldsymbol{\omega}}^*(a|s, \boldsymbol{\omega})$ or value $\boldsymbol{Q}(s, a, \boldsymbol{\omega})$ using MODF or PEDA
 4: Approximate the prior preference distribution $P(\boldsymbol{\omega}|\mathcal{D})$
 5:
 6: ### Adaptation Phase
 7: **Input:** Demonstration set $\mathcal{B}$, and conservatism parameter $\alpha$
 8: Initialize $\mathcal{N}(\boldsymbol{\mu}, \boldsymbol{\sigma I})$ based on $P(\boldsymbol{\omega}|\mathcal{D})$
 9: **for** each interaction **do**
10:     Sample preference $\boldsymbol{\omega}$ from $\mathcal{N}(\boldsymbol{\mu}, \boldsymbol{\sigma I})$
11:     Update $\boldsymbol{\mu}, \boldsymbol{\sigma}$ through gradient descent (Eq. (7))
12: **end for**
13: **if** no constraint **then**
14:     $\tilde{\boldsymbol{\omega}}_a = \boldsymbol{\mu}/|\boldsymbol{\mu}|$
15: **else**
16:     Compute $\boldsymbol{b}$ based on Eq. (10)
17:     $\tilde{\boldsymbol{\omega}}_a = \boldsymbol{b}/|\boldsymbol{b}|$
18: **end if**
19: **Output:** Policy $\hat{\pi}_{\boldsymbol{\omega}}^*(\cdot|\cdot, \tilde{\boldsymbol{\omega}}_a)$ for decision making

---

is generally more desirable. However, the sparsity metric is not applicable in our setting. This is because, given that the number of adapted policies is fixed, an increase in utility and Hypervolume performance can result in a worse sparsity metric. Moreover, better sparsity metrics encourage a lower diversity of adapted policies, which diverges from our intended goal.

### A.5 Implementation Details

The pseudocode for the training and adaptation process is provided in Algorithm 1.

**Behavioral Preference Approximation Scheme in Constrained Settings**   In constrained settings, the vector reward is defined as $\tilde{\boldsymbol{r}} = [\boldsymbol{r}_t, -\boldsymbol{c}_t]$ and dataset $\mathcal{D}$ is augmented to $\hat{\mathcal{D}}$. For behavioral preferences in $\hat{\mathcal{D}}$, we utilize $\tilde{\boldsymbol{\omega}} = U(\tau)/\|U(\tau)\|_1$ as behavioral preferences for all transitions in trajectory $\tau$. Here, $U(\tau) = [\boldsymbol{R}_1, ..., \boldsymbol{R}_N, \boldsymbol{C}_1^{\max} - \boldsymbol{C}_1, ..., \boldsymbol{C}_K^{\max} - \boldsymbol{C}_K]$, where $\boldsymbol{R}_i$ and $\boldsymbol{C}_i$ represent the accumulations of the $i^{\text{th}}$ reward and cost of $\tau$, respectively, and $\boldsymbol{C}_i^{\max}$ is the maximum of $\boldsymbol{C}_i$ among all trajectories. The formulation of $\tilde{\boldsymbol{\omega}}$ aims to approximate the normal vector of the Pareto front that consists of $(\boldsymbol{R}_1, ..., \boldsymbol{R}_N, -\boldsymbol{C}_1, ..., -\boldsymbol{C}_K)$.

**Preference Distribution Offline Adaptation (PDOA)**   We utilize the policy and value function obtained by MODF and PEDA to calculate the TD reward and action likelihood reward in Eq. (7). For MODF, except the state-action value function $\boldsymbol{Q}(s, a, \boldsymbol{\omega}) = Q_1(s, a, \boldsymbol{\omega}), ..., Q_N(s, a, \boldsymbol{\omega})$ obtained during training, we additionally train a state value function $\boldsymbol{V}(s, \boldsymbol{\omega}) = \mathbb{E}_{\pi(a|s, \boldsymbol{\omega})}[\boldsymbol{Q}(s, a, \boldsymbol{\omega})]$ to derive TD reward. As for the action likelihood reward, the action likelihood of the diffusion policy model is intractable but can be approximated as $\log \hat{\pi}_{\boldsymbol{\omega}}^*(a|s, \boldsymbol{\omega}) \approx \mathbb{E}_{i \sim \mathcal{U}, \epsilon \sim \mathcal{N}(\mathbf{0}, \boldsymbol{I})}\|a - \hat{a}^0\|_2^2$ [Kang et al., 2024], where $\hat{a}^0$ is reconstruction of $a^i$ and $a^i$ is obtained by corrupting $a$ with noise $\epsilon$. For PEDA, which does not involve value functions, we disregard the TD reward when calculating the adaptation loss. The prior preference distribution $P(\boldsymbol{\omega}|\mathcal{D})$ in Eq. (7) is approximated with a Gaussian distribution $\mathcal{N}(\boldsymbol{\mu}_{\mathcal{D}}, \boldsymbol{\sigma}_{\mathcal{D}})$, where the expectation $\boldsymbol{\mu}_{\mathcal{D}}$ and standard deviation $\boldsymbol{\sigma}_{\mathcal{D}}$ are estimated on behavioral preferences of the training dataset $\mathcal{D}$. The weight $\eta$ of the regularization term $(\|\boldsymbol{\mu}\|_1 - 1)^2$ in Eq. (7) is set to $1.0$.

During adaptation, we model the preference distribution using a Gaussian model and perform gradient updates on this distribution according to Eq. (7). For each target, the number of gradient updates is set to 1000, with 64 preferences sampled from the distribution for each gradient update. All samples in the demonstration set are used for gradient updates within one batch. We use the Adam optimizer with a learning rate of $0.05$. The conservatism weight $\alpha$ in Eq. (10) is set to $1.0$ for MORL tasks

and 0.7 for safe RL and CMORL tasks. The weight of the TD reward in Eq. (6) is set to 0.01 for PDOA [MODF].

**MORL and Safe RL Algorithms**  For **MODF**, we use the original implementation in `https://github.com/qianlin04/PRMORL`. We remove the Regularization Weight Adaptation component in the original MODF since it requires additional online interactions during evaluation. We follow the default hyperparameters except for the regularization weight, which is set to 200 for MORL and CMORL tasks and 20 for safe RL tasks. For **PEDA**, we use the original implementation in `https://github.com/baitingzbt/PEDA`. PEDA utilizes a linear regression model to predict target vector returns given target preferences. For D4MORL datasets, it fits this model using Expert datasets. In DSRL and CMO datasets, which contain suboptimal samples, we construct a Pareto set that consists of all near-undominated trajectories with a small tolerance (5%) and use it to fit the linear regression model. For **Prompt-MODT**, we implement it based on the MODT algorithm in PEDA by incorporating prompts into the decision transformer. The target vectors predicted by the above linear regression model are used as the initialized return-to-go for Prompt-MODT. Besides, to apply Prompt-MODT, we need to transform the multi-objective problem into a multi-task problem, for which we divide trajectories with different behavioral preferences into separate tasks with a preference interval of 0.05. For example, trajectories with behavior preferences ranging from $[0.00, 1.00]$ to $[0.05, 0.95]$ are classified as task 1, and trajectories with behavior preferences ranging from $[0.05, 0.95]$ to $[0.10, 0.90]$ are classified as task 2. For **BC-Finetune**, we model the policy using a Gaussian model and employ behavioral cloning through MSE loss. During adaptation, behavior cloning is performed on the demonstration set with 0.01 learning rate and 1000 gradient update steps.

**Computer Resources**  The training and testing were conducted on 1 NVIDIA GeForce RTX 3090 GPU. The testing utilizes 5 CPU threads to simultaneously collect test data from multiple environments. The total time for training and testing does not exceed 10 hours. The memory usage for a single run depends on the size of the dataset used, but generally does not exceed 10 GB.

## A.6    Additional Results

### A.6.1    Experimental Results on D4MORL Expert Datasets for MORL

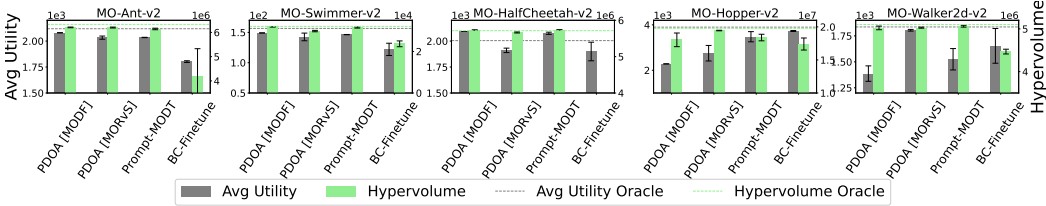

Figure 8: Average utility and Hypervolume performance of all algorithms on D4MORL Expert datasets.

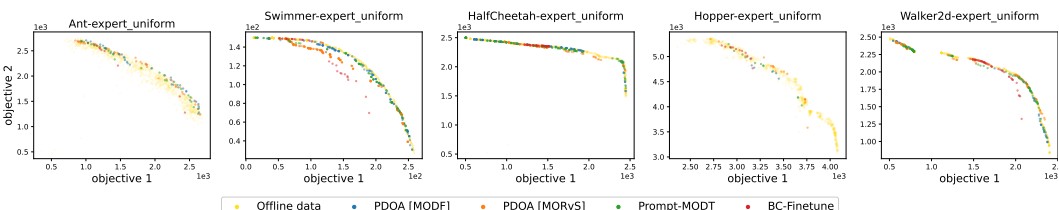

Figure 9: Pareto fronts of different algorithms on D4MORL Expert datasets. Each point represents the expected vector return of an adapted policy for a specific unknown target preference.

We present the average utility and Hypervolume performance, Pareto fronts, and the difference between real target preferences and adapted preferences on the D4MORL Expert datasets in Figure 8,

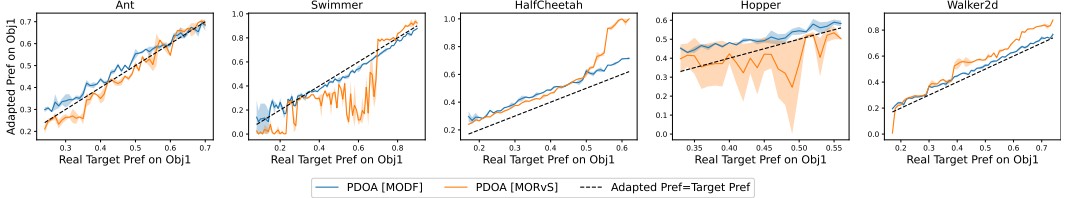

Figure 10: The comparison between real target preferences and adapted preferences on D4MORL Expert datasets.

9, and 10, respectively. In Figure 8, our method demonstrates overall competitive performance in terms of average utility and Hypervolume metrics. Figure 9 illustrates that our method achieves a broader Pareto front, indicating a high diversity of adapted policies. Figure 10 shows that our method can obtain expected behaviors due to the consistency between our adapted preferences and real target preferences. These results are consistent with those obtained on the Amateur Datasets in Section 5.1, which demonstrates that our framework achieves consistent advantages across datasets of varying quality.

## A.7 Experimental Results on Additional DSRL Tasks for Safe RL

Figures 11 and 12 show the performance of MODF and MORvS under various target preferences, which demonstrate the capability of these MORL algorithms in obtaining policies that satisfy different safety thresholds. Figure 13 illustrates the normalized cost and reward for additional DSRL tasks, including CarCircle, DroneCircle, CarRun, and DroneRun. The results are consistent with those in Section 5.2, demonstrating that MORL methods can effectively learn a set of policies that meet various safety thresholds, and that PDOA [MODF] can achieve very few constraint violations.

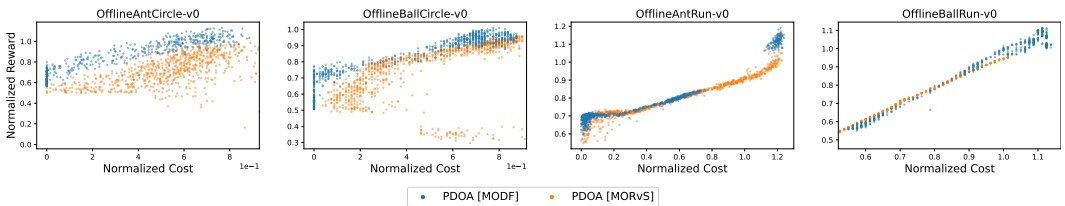

Figure 11: The normalized cost and normalized reward of policies with various preferences obtained by our framework.

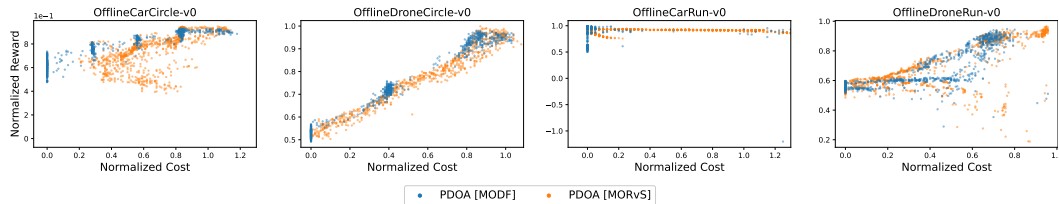

Figure 12: The normalized cost and normalized reward of policies with various preferences obtained by our framework on additional tasks.

## A.8 Comparison with Offline Imitation Learning Baseline

Applying offline IL to our problem faces several challenges: 1) Offline IL requires complete policy training for each adaptation, which is resource-intensive and time-consuming. 2) Our experiments

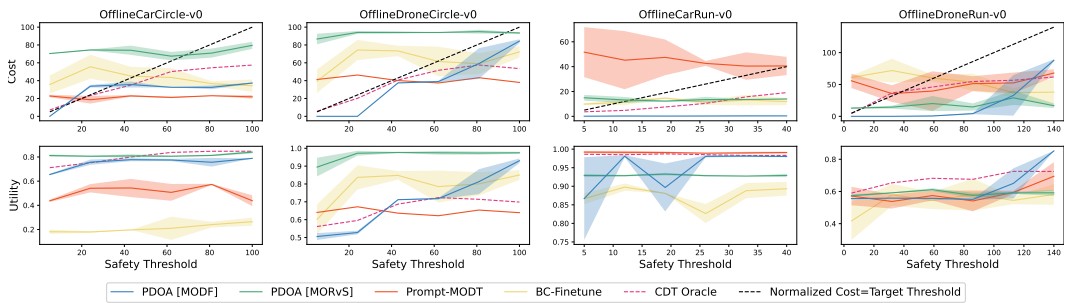

Figure 13: The adapted policies' normalized cost and normalized reward of all algorithms under various safety thresholds on additional tasks.

involve a limited number of demonstrations (128 transitions per target preference) compared to the millions of samples in the offline dataset, leading to significant demo sufficiency and data imbalance.

We apply an OIL baseline DWBC [Xu et al., 2022b] in our MORL and safe RL experiments and present the results in Figure 14 and 15, where we limit the number of tested preferences to 10 in the D4MORL tasks because the original evaluation involves dozens of target preferences per task, and training a policy for each target preference using DWBC would be too time-consuming. In both MORL and safe RL environments, DWBC demonstrates unstable performance. It is competitive in tasks such as MO-Ant, MO-Swimmer, but performs poorly in other tasks, including MO-Walker2d. It even struggles to learn effective policies in some cases, such as MO-Hopper and AntCircle. These issues can be attributed to demo sufficiency and data imbalance. Additionally, DWBC experiences high constraint violations in most safe RL tasks, as it does not incorporate safety considerations into its policy learning.

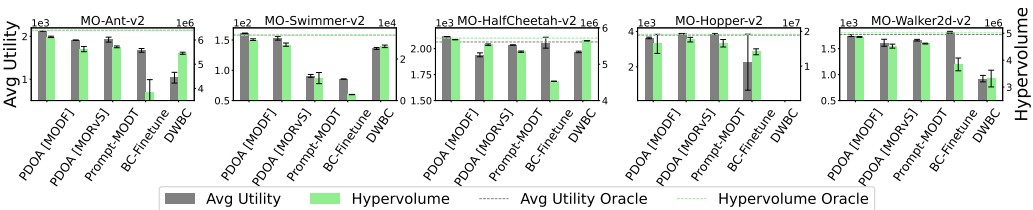

Figure 14: Comparative results with offline IL baseline on MORL tasks.

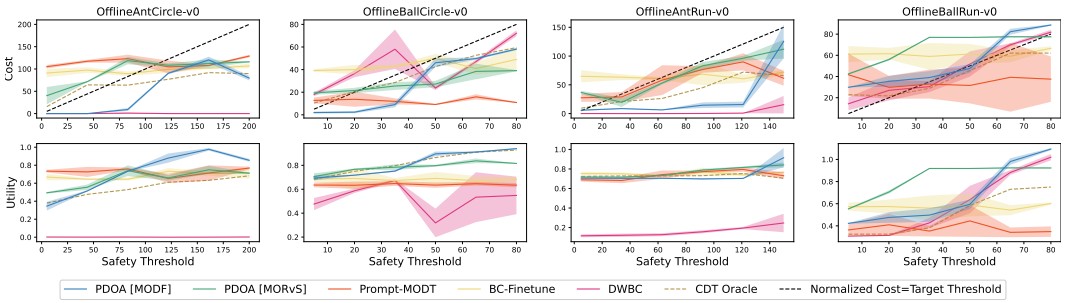

Figure 15: Comparative results with offline IL baseline on safe tasks.

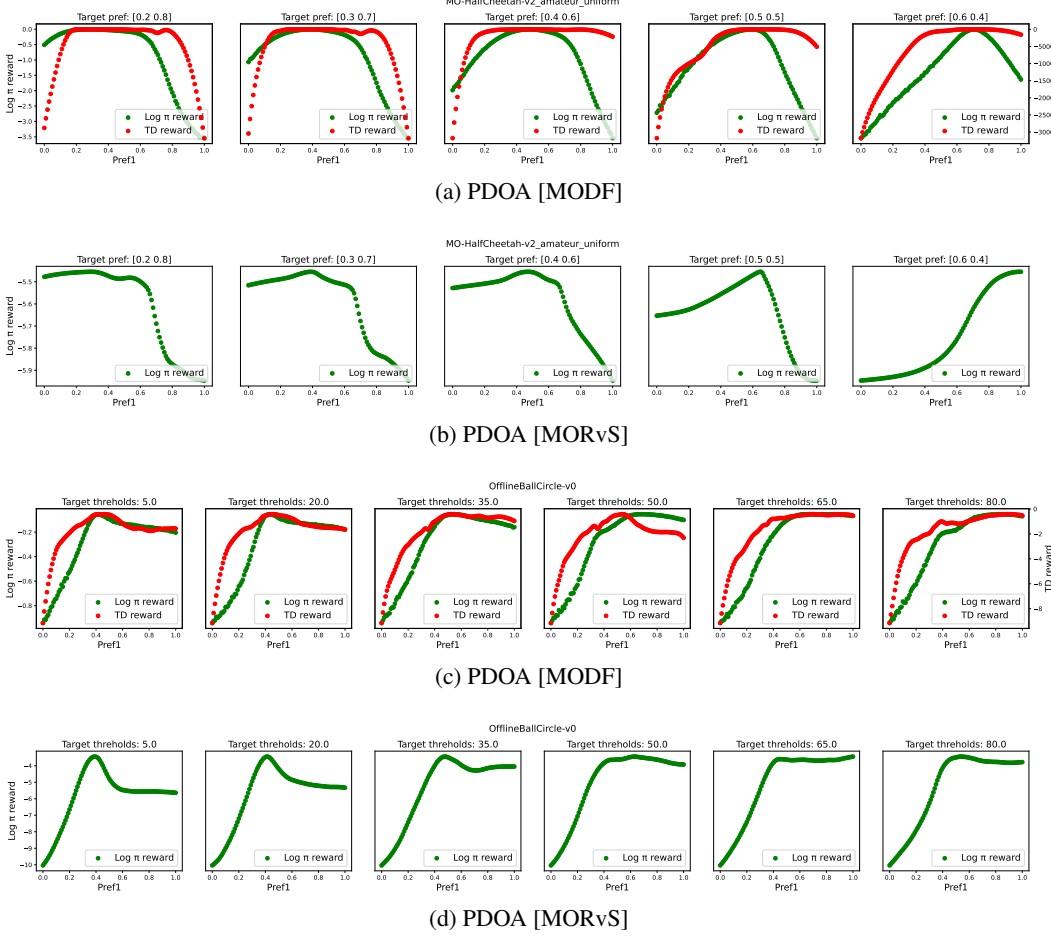

Figure 16: The relationship between TD reward, action likelihood reward, and preference. Pref1 represents the first dimension of preference.

## A.9 Relationship between TD Reward, Action Likelihood Reward and Preference

For each target preference, we traverse all possible adapted preferences with a small interval and compute the corresponding TD reward and action likelihood reward. The results are presented in Figure 16, where we observe that the adapted preferences corresponding to the highest TD reward and action likelihood reward align with the target preferences. Additionally, as constraints are relaxed, the preference weights allocated to the constrained targets decrease. This demonstrates that the TD reward and action likelihood reward in our algorithm are strongly consistent with the target preferences and safety thresholds.

## A.10 Licenses

- D4MORL dataset and PEDA code [Zhu et al., 2023]: The MIT License, `https://github.com/baitingzbt/PEDA`

- Datasets and codes of DSRL, OSRL and FSRL [Liu et al., 2023b]: All datasets are licensed under the Creative Commons Attribution 4.0 License (CC BY 4.0), and code is licensed under the Apache 2.0 License. `https://github.com/liuzuxin/DSRL`, `https://github.com/liuzuxin/OSRL`, `https://github.com/liuzuxin/FSRL`

- MuJoCo: Apache 2.0 License, `https://github.com/google-deepmind/mujoco`

### A.11 Broader Impacts

Our research introduces a novel framework for addressing constrained multi-objective reinforcement learning (CMORL), which is widely applicable to many real-world scenarios, such as autonomous driving, healthcare, where behavioral preferences and safety criteria are difficult to define precisely. These approaches liberate researchers from the labor required to align preferences and safety thresholds. Besides, the offline training paradigm eliminates the expenses and dangers associated with online exploration. For instance, in autonomous driving, online interactions with the environment have the risk of accidents and injuries; however, our method can mitigate this risk by learning from a pre-recorded driving dataset generated by safe and target behavior policies.

Nonetheless, our method may not be suitable for domains requiring rapid responses and frequent updates due to the additional computational load and latency caused by gradient updates during adaptation. Besides, providing personalized services with this framework necessitates a certain amount of user data as demonstrations, raising concerns about privacy breaches and data misuse. Despite the risks and challenges mentioned, we believe that CMORL holds great promise for automating and enhancing sequential decision-making in highly impactful domains. Additional work is required to make this framework robust enough for application in multi-objective and safety-critical scenarios.

