# OpenReview forum: "An Offline Adaptation Framework for Constrained Multi-Objective Reinforcement Learning"
_NeurIPS.cc/2024/Conference — NeurIPS 2024 poster_

### Official Review · Reviewer_TEUx · 2024-06-26

**Soundness:** 3
**Presentation:** 3
**Contribution:** 3
**Rating:** 6
**Confidence:** 4

**Summary:**

This paper presents a novel framework for multi-objective RL  that eliminates the need for handcrafted target preferences. Instead, it uses demonstrations to implicitly indicate the preferences of expected policies. The proposed offline adaptation framework can also handle safety-critical objectives by utilizing safe demonstrations, even when safety thresholds are unknown. The authors provide empirical results showing that the framework effectively infers policies that align with real preferences and meet the constraints implied by the demonstrations.

**Strengths:**

(1) Interesting topics: Developing a framework for multi-objective reinforcement learning that does not rely on handcrafted target preferences is very interesting and potentially useful for real-world applications. Additionally, the framework uses demonstrations to implicitly indicate the preferences of expected policies, which can also be extended to meet constraints on safety-critical objectives using safe demonstrations.

(2) Extensive experiment evaluation: The authors provided extensive experiment results including those on offline safe RL benchmark, offline RL benchmark, and multi-objective RL benchmark. These results show the flexibility and effectiveness of the proposed method.

**Weaknesses:**

(1) Preference is closely related to the reward function definition. How does this definition compare to other metrics such as the ones based on the states/state-action/latent variable distribution? For some tasks, there exists some correlation between reward functions, and this will potentially affect the preference estimation results.

For other potential weaknesses, please see my questions.

**Questions:**

(1) Preference is closely related to the reward function definition, and correlation among reward functions will result in preference calculation issues. For example, what if $r_1 = 0.5 r_2 = 0.2 r_3$, and $r_4$ is independent to $r_1$? In this setting, we would get a small preference mismatch score between two demonstrations even if they have significant differences in the reward returns for $r_4$. Do you have any methods to deal with this?

(2) In section 4.2, the first step of your method is to learn a set of policies that respond to various preferences during training. How do you ensure that the learned policy is subject to specific preferences?

(3) Based on the description in section 4.3, are High-reward-high-cost and low-reward-low-cost demonstrations categorized as similar behaviors?

(4) How does the proposed method perform in other DSRL datasets/tasks such as Car-Circle and Drone-Circle?

(5) Where are CDT results discussed in section 5.2? I did not them in Figures 3 and 4.

(6) How do your method and problem formulation relate to multi-constraint safe RL since multiple constraints can also be viewed as multiple objectives? Some related works:

[1] Kim, D., Lee, K., & Oh, S. (2024). Trust region-based safe distributional reinforcement learning for multiple constraints. Advances in neural information processing systems, 36.

[2] Yao, Y., Liu, Z., Cen, Z., Huang, P., Zhang, T., Yu, W., & Zhao, D. (2023). Gradient Shaping for Multi-Constraint Safe Reinforcement Learning. arXiv preprint arXiv:2312.15127.

[3] Guan, J., Shen, L., Zhou, A., Li, L., Hu, H., He, X., ... & Jiang, C. (2024). POCE: Primal Policy Optimization with Conservative Estimation for Multi-constraint Offline Reinforcement Learning. In Proceedings of the IEEE/CVF Conference on Computer Vision and Pattern Recognition (pp. 26243-26253).

**Limitations:**

Yes, the authors addressed the limitations.

---

> ### Author Rebuttal · Authors · 2024-08-06
>
> We thank the reviewer for all the valuable comments. Please refer to the point-to-point responses below.
>
> **W1 & Q1: Impact of reward correlation on preference match**
>
> We might not fully understand the reviewer's concern about reward correlation, so we would appreciate further clarification if we have misunderstood. The concern seems to be that two policies with similar preferences might behave differently on one objective and thus generate different demonstrations because their preferences assign low weights to this particular objective.
>
> Nevertheless, our approach does not suffer from this issue because practically it matches a demonstration to a specific preference based on its behavioral similarity to the learned policies (as indicated by the item $\log \hat{\pi}^*_{\omega}(a_t|s_t,\omega)$ in Eq. (7)), rather than directly comparing the real preferences of demonstrations to those of trajectories in the dataset. As a result, even if two demonstrations are generated by policies with similar preferences, they will lead to distinct behaviors within our framework.
>
>
>
> **Q2: How ensure the learned policy to be subject to preferences**
>
> The policy with a specific preference is expected to achieve high scalar utility, which is a linear combination of the preference weights and multiple rewards in our work. To accomplish this, MODF estimates the value vector $\boldsymbol{Q}(s,a,\omega)$ under different preferences and updates the policies to maximize $\omega \boldsymbol{Q}^T$. In contrast, PEDA falls into the return-conditioned paradigm, where it estimates the return vectors of expert policies for each preference and uses both the preference labels and these return vectors as decision conditions during behavior cloning. This approach enables it to approximate the expert's performance for each preference by conditioning on the specific preference and the predicted expert return vector.
>
>
>
> **Q3. Difference between High-reward-high-cost and low-reward-low-cost demonstrations**
>
> High-reward-high-cost demonstrations are generated by policies with a high preference weight on rewards and a low preference weight on constrained objectives (i.e., $r=-c$). Conversely, low-reward-low-cost demonstrations are generated by policies with the opposite preference (low weight on rewards and high weight on constrained objectives). Thus, these demonstrations should be associated with completely different behaviors.
>
> **Q4. Performance on other DSRL tasks**
>
> Figure 13 illustrates the results on additional DSRL tasks, including CarCircle, DroneCircle, CarRun, and DroneRun. The results are consistent with those in Section 5.2, showing that our method achieves fewer constraint violations and higher rewards compared to baselines and approaches the CDT oracle's performance.
>
> **Q5. Lack of CDT results in Figure 4**
>
> We appreciate the reviewer pointing out this issue. In fact, the lines labeled 'MODF oracle' in Figure 4 and Figure 13 represent the performance of CDT. We will correct these legends in the future version.
>
>
>
> **Q6. Connection to multiple constraint setting**
>
> As discussed in Section 4.3, multiple constraints can be viewed as additional rewards. This approach allows the original constraint problem to be equivalently converted into an MORL problem, which can then be solved through offline MORL adaptation by assigning appropriate preference weights to these constrained objectives.
>
> Existing research on multi-constraint RL typically explores the correlation between multiple constraints from gradient perspectives [1,2]. Some works [3] attempt to meet different types of constraints simultaneously. In contrast, our approach, from the policy diversity perspective, aims to find appropriate policies for various safety requirements and thus places less focus on constraint correlation for a specific threshold. Nevertheless, previous studies on multiple constraint RL will provide important insights into managing potential correlations between multiple objectives during preference-conditioned policy learning, thereby improving overall performance.
>
> Additionally, we will supplement the related studies and discussions in our future version.
>
>
>
> [1] Kim, D., Lee, K., & Oh, S.. Trust region-based safe distributional reinforcement learning for multiple constraints. NeurIPS 2024.
>
> [2] Yao, Y., Liu, Z., Cen, Z., Huang, P., Zhang, T., Yu, W., & Zhao, D.. Gradient Shaping for Multi-Constraint Safe Reinforcement Learning. L4DC 2024.
>
> [3] Guan, J., Shen, L., Zhou, A., Li, L., Hu, H., He, X., ... & Jiang, C.. POCE: Primal Policy Optimization with Conservative Estimation for Multi-constraint Offline Reinforcement Learning. CVPR 2024.

---

> > ### Comment · Reviewer_TEUx · 2024-08-09
> >
> > Thanks for the additional explanation. All my concerns and questions have been successfully addressed. I have raised my score from 5 to 6 in favor of acceptance.

---

> ### Author Response · Authors · 2024-08-10
>
> Thanks again for all the insightful comments and suggestions. They are helpful in improving our paper.

---

### Official Review · Reviewer_58tv · 2024-07-17

**Soundness:** 4
**Presentation:** 4
**Contribution:** 3
**Rating:** 6
**Confidence:** 3

**Summary:**

The paper presents a new multi-objective RL setup to learn a policy offline that can adapt to unknown target preference online, utilizing posterior estimates of the true target preference.

**Strengths:**

1. The paper is very clearly written.
2. The base policy is reasonable (decision transformer and diffusion policy).
3. Experiment and figures are easy to follow.
4. The methodology is built on solid prior work and the extensions are easy to understand.

**Weaknesses:**

After carefully thinking through this setup, I'm a bit worried about the core contribution of this paper.

There are several major challenges to offline learning of CMO-MDP:
1. Policy learning conditioned on preference vector (solved by previous works and used in this paper) (MODF, MORvS)
2. Trajectory distributions shift under the optimal policy and approximate policy, leading to failures of estimating preference vector w (solved by [Ghosh et al., Hong et al., 2022], Eq 6, used in this paper)
3. Preference vector online estimation (through posterior update) (Eq 5) (this is a straightforward application), and the actual update is Eq 7.

It seems to me that the paper's core contribution is:
1. Proposed a new problem.
2. Proposed a good preference target vector estimation (Eq 7).
3. Cast constrained problem as unconstrained and added a conservative estimate to the preference vector so that constraints are not violated during adaptation.

I'm not fully able to determine how significant these contributions are.

**Questions:**

For Eq 5, can the author comment on -- what is the estimation quality of the target preference policy if the sampling policy pi-hat-star-w is sub-optimal? I can break this question down to two parts:
- How does the sub-optimality of the approximate policy to optimal policy affect the estimation of w?
- How does the estimation quality of w affect the convergence of the approximate policy to the true optimal policy?

**Limitations:**

The authors addressed the limitation.

---

> ### Author Rebuttal · Authors · 2024-08-06
>
> We thank the reviewer for all the valuable comments. Please refer to the point-to-point responses below.
>
> **W1: About contribution**
>
> First, we would like to clarify that our primary contribution is not to develop a new MORL algorithm parallel to MODF or MORvS but to propose a flexible and minimalist framework designed to accommodate existing MORL algorithms and address new problems that the original algorithms cannot solve.
>
> Regarding the approximation technique in Eq. (6), we consider it more of an effective and replaceable trick rather than a critical component of the work. Because, there are several methods that could fulfill this role, such as explicitly modeling dynamics and stationary distributions through model-based methods or DICE techniques. We selected Ghosh's method due to its simplicity and effectiveness, as it does not require additional mechanisms that could complicate our framework and hinder its generalization.
>
> Furthermore, while the Bayesian formulation is intuitive, its application to MORL adaptation problems is relatively novel. It serves as the foundation of our solution and offers a new perspective for MORL research. Finally, we emphasize the significance of incorporating safe RL into our framework. The equivalence relation we propose is not only intuitive but also provides a unified perspective on both safe RL and MORL.
>
>
>
> **About 'optimal policy'**
>
> Obtaining and evaluating the real optimal policy is challenging, especially in an offline setting where the quality of the learned policies is significantly influenced by the dataset. We use the term 'optimal policy' as a theoretical bridge to bypass the suboptimality of the learned policy and simplify the analysis. Despite this discrepancy, our approach achieves meaningful adaptation. This is because, in practice, we select a well-chosen preference that ensures the learned policy’s behavior with this preference closely aligns with the demonstrations, and the dynamics estimation based on these demonstrations remains relatively accurate.
>
> **How does the sub-optimality of the approximate policy to optimal policy affect the estimation of w?**
>
> We can easily find that the KL divergence between the real posterior $P(\omega|B,D)$ and the approximate one $\hat P(\omega|B,D)$ in Eq. (5) is bounded by the KL divergence between the policy, state distribution, and dynamics in the real environment and their empirical counterparts. i.e.,
> $$
> D_{KL}(\hat P(\omega|B,D), P(w|B,D))\leq E_{\hat\pi,\hat M}[D_{KL}(d_{\hat\pi}(s), d_{\pi^*}(s))+D_{KL}(\hat \pi(a|s),\pi^*(a|s))+D_{KL}(\hat P(s',r|s,a),P(s',r|s,a))]+K.
> $$
> Thus, if the learned policies deviate significantly from the optimal policies, the estimation of $\omega$ will be adversely affected.
>
> **How does the estimation quality of w affect the convergence of the approximate policy to the true optimal policy?**
>
> The adapted preference $\omega$ does not involve in policy training but still impacts the performance discrepancy between the adopted policy and the optimal policy. Even if we could obtain optimal policies for all preferences, poor performance could still arise if the predicted target preference deviates significantly from the actual one. This is because a policy might perform well under the predicted preference but diverge from the optimal policies with the real preference.
>
>
>
> [1]. Ghosh, D., Ajay, A., Agrawal, P., & Levine, S. (2022, June). Offline rl policies should be trained to be adaptive. In *International Conference on Machine Learning* (pp. 7513-7530). PMLR.

---

> ### Author Response · Authors · 2024-08-13
>
> Thanks for insightful suggestions. If there are still confusions, please let us know. We would be happy to provide further clarifications.

---

### Official Review · Reviewer_ZqLH · 2024-07-18

**Soundness:** 2
**Presentation:** 3
**Contribution:** 3
**Rating:** 6
**Confidence:** 3

**Summary:**

This paper addresses the issue of handcrafted target preferences in multi-objective RL (MORL) by proposing an offline adaptation framework called Preference Distribution Offline Adaptation (PDOA), which implicitly learns the preferences from a small number of demonstrations. Specifically, PDOA involves two steps: (i) It learns a set of policies for various preferences during training using an off-the-shelf MORAL algorithm; (ii) It then adapts a distribution of target preferences based on the demonstrations at deployment. To update the posterior of target preferences in a tractable manner, PDOA adopts multiple approximation schemes for both unconstrained and constrained settings. Moreover, in the constrained setting, to address the discrepancy between the true preference and the estimated one, PDOA also proposes to use a conservative estimate with the help of CVaR. The PDOA is evaluated on multiple MOD4RL locomotion tasks and has strong performance in terms of average utility and hypervolume. Empirical studies on the number of demonstrations and the conservatism parameter are also provided.

**Strengths:**

- This paper identifies and addresses the issue of target preference estimation by offline preference adaptation, which is a novel setting in the literature of MORL.
- The idea of rethinking target preferences from an Bayesian inference perspective is quite interesting. The overall approach looks reasonable (despite some concerns listed below).
- The experiments are done on multiple D4MORL tasks, and the results in terms of utility and hypervolume are quite promising.
- The ablation studies are quite helpful in understanding the robustness of PDOA.

**Weaknesses:**

- One issue is the lack of stronger offline imitation learning baselines. If we already have a number of demonstrations of the target preferences at test time, then one reasonable solution is to view this as a single-objective problem and then apply an off-the-shelf offline imitation learning (IL) algorithm with these demonstrations, such as DWBC [Xu et al., 2022] and DemoDICE [Kim et al., 2022], which both focus on offline imitation with a small number of expert demonstrations along with other sub-optimal trajectories. In the setting of PDOA, one can simply do offline IL by using $\mathcal{B}_\omega$ as the expert demonstrations and $\mathcal{D}$ as the sub-optimal samples.
- Another concern is on approximation of the posterior in (6). While I can get the intuition that in-distribution samples tend to have higher $r^{TD}$, the surrogate in (6) appears to be quite heuristic and not very principled. Specifically, the $\pi^*$ in the original posterior expression could be different from the policy (whose Q and V are used in Equation 6) learned by MORL at the first stage. More theoretical justification of this approximation is needed.
- For the constrained setting, there is an additional parameter $\alpha$ to determine to strike a balance between constraint violation and achieved utility. However, my concern is that $\alpha$ is a parameter to be used at test time, and hence there is very little one can do in terms of parameter tuning. While Figure 6 did provide a study on the influence of $\alpha$, we can see that this value does matter a lot in the balance between constraint violation and utility. As a result, this could make PDOA less attractive in practice.

[Xu et al., 2022] Haoran Xu, Xianyuan Zhan, Honglei Yin, and Huiling Qin, “Discriminator-Weighted Offline Imitation Learning from Suboptimal Demonstrations,” ICML 2022.

[Kim et al., 2022] Geon-Hyeong Kim, Seokin Seo, Jongmin Lee, Wonseok Jeon, HyeongJoo Hwang, Hongseok Yang, and Kee-Eung Kim, “DemoDICE: Offline Imitation Learning with Supplementary Imperfect Demonstrations,” ICLR 2022.

**Questions:**

Please see the comments above. Some additional questions:
- How many trajectories (in the demonstration set) are needed for target preference adaptation in the main experiments (Figures 1-2)?
- In Equation 5, during the preference estimation phase, the dataset $\mathcal{D}$ only takes part in the prior $P(\omega| \mathcal{D})$. Under PDOA, it could be possible to leverage the information from $\mathcal{D}$ to further improve the accuracy of the target preference. Can the authors comment on this?

**Limitations:**

The paper discusses one limitation about the need for human expertise in constructing demonstrations in Section 6.

---

> ### Author Rebuttal · Authors · 2024-08-06
>
> We thank the reviewer for all the valuable comments. Please refer to the point-to-point responses below.
>
> **W1: Lack of offline IL baselines**
>
> Applying offline IL to our problem faces several challenges: 1) Offline IL requires complete policy training for each adaptation, which is resource-intensive and time-consuming. 2) Our experiments involve a limited number of demonstrations (128 transitions per target preference) compared to the millions of samples in the offline dataset, leading to significant demo sufficiency and data imbalance.
>
> Nevertheless, we still apply an OIL baseline DWBC [1] in our MORL and safe RL experiments and present the results in the PDF attached to the global response. We use the same evaluation protocol as in the original study, except that we limite the number of tested preferences to 10 in the D4MORL tasks. This limitation is necessary because the original evaluation involves dozens of target preferences per task, and training a policy for each target preference using DWBC would be too time-consuming.
>
> In both MORL and safe RL environments, DWBC demonstrates unstable performance. It is competitive in tasks such as MO-Ant, MO-Swimmer, and CarCircle, but performs poorly in other tasks, including MO-Walker2d, and DroneRun. It even struggles to learn effective policies in some cases, such as MO-Hopper and AntCircle. These issues can be attributed to demo sufficiency and data imbalance. Additionally, DWBC experiences high constraint violations in most safe RL tasks, as it does not incorporate safety considerations into its policy learning.
>
> Finally, we would discuss the related works and consider to supplement this results in the future version.
>
>
>
> **W2: Justification of TD-reward surrogate in Eq. (6) and discrepancy between the theoretical optimal policy and the one learned by MORL**
>
> Admittedly, the use of TD-error approximation from Ghost's paper [2] is heuristic and based on empirical observations (as shown by the consistency of the TD term and real preference in Figure 14 of Appendix A.9). The related analysis involves the generalization of Q-value neural networks and its relationship to dynamics uncertainty, which makes it extremely complex. Alternative methods like model-based approaches or DICE methods could potentially offer more explicit dynamics modeling and theoretical insights, though they would also complicate the overall approach. Nevertheless, exploring these alternatives could provide more reasonable solutions and is a direction for future research.
>
> Regarding the discrepancy between the learned policy $\pi$ and the optimal policy $\pi^*$, it is evident that the KL divergence between the real posterior and the approximate one in Eq. (5) is bounded by the KL divergence between the policies, state distributions, and dynamics in the real and empirical environments, i.e.,
> $$
> D_{KL}(\hat P(\omega|B,D), P(w|B,D))\leq E_{\hat\pi,\hat M}[D_{KL}(d_{\hat\pi}(s), d_{\pi^*}(s))+D_{KL}(\hat \pi(a|s),\pi^*(a|s))+D_{KL}(\hat P(s',r|s,a),P(s',r|s,a))]+K.
> $$
> Therefore, as the learned policy approaches the optimal policy, the approximation gap decreases. However, the analysis of policy optimality for diffusion-based regularized methods in MODF and supervised RL methods in PEDA is complex and limited in the related works (as far as we know). Thus, we leave a detailed theoretical analysis for future research.
>
> Besides, we use the 'optimal policy' just as a theoretical bridge to simplify the analysis. Despite this discrepancy, our approach can still achieves meaningful adaptation because what we did is to select a well-chosen preference that ensures the learned policy’s behavior with this preference closely aligns with the demonstrations. Practically, this process does not involve the policy optimality directly.
>
>
>
> **W3: Conservative parameter $\alpha$**
>
> First, since the parameter $\alpha$ is used only in the deployment phase and each adaptation involves relatively few samples, fine-tuning $\alpha$ is manageable. This can be accomplished with a few iterations of adaptation, which is feasible in many real-world applications.
>
> Moreover, Figure 6 demonstrates that our approach maintains relatively few constraint violations even when the conservatism mechanism is removed (i.e., $\alpha=1$). This is because the equivalence relationship between constraint RL and MORL is key to solving constraint problems, while conservatism with $\alpha$ serves as an enhancement based on this equivalence. Thus, a roughly designed $\alpha$ will not significantly undermine our approach.
>
> **Q1: Number of trajectories in the demonstration set**
>
> We select 2 trajectories from the original dataset and then randomly sample 128 transitions from these trajectories to construct a demonstration set for each target preference or safety threshold. For further details, please see Appendix A.3.
>
>
>
> **Q2: Potential Improvement from Extended Utilization of Dataset $D$**
>
> Except for $P(\omega|D)$, dataset $D$ also appears in the term $P(B_{\omega_g}|\omega,D)$ in Eq. (5), which is approximated using policy and dynamics estimation. This means that $D$ not only provides prior preference distribution information but also plays a role in calculating $r^{TD}_\omega$ and $\log \pi$ in adaptation (i.e., Eq. (7)), as it is required when learning preference-conditioned models during the first phase.
>
> Nevertheless, I agree that further leveraging this information is possible. For instance, measuring the similarity between demonstrations and samples in $D$ somehow could provide additional insights for better inference.
>
> [1]. Xu, H., Zhan, X., Yin, H., & Qin, H.. Discriminator-weighted offline imitation learning from suboptimal demonstrations. ICML 2022.
>
> [2]. Ghosh, D., Ajay, A., Agrawal, P., & Levine, S.. Offline rl policies should be trained to be adaptive. ICML 2022.

---

> ### Comment · Reviewer_ZqLH · 2024-08-12
> **Thanks for the response**
>
> Thank the authors for the detailed response. My concerns about the baselines and the demonstration set have been nicely addressed. Some follow-up comments / questions:
>
> - About W2: My question about the justification for $r^{TD}$ remains unresolved. To justify the use of (6), one would need to roughly show that $r^{TD}$ is generally larger when the posterior is larger. Somehow I do not see why the explanation in the rebuttal is relevant. Please let me know if I missed anything.
>
> - About W3: It is mentioned that “...a roughly designed $\alpha$ will not significantly undermine our approach.” However, in Figure 6(c)-(d), it appears that a difference of 0.2 in $\alpha$ can indeed have a significant impact on both the cost and the utility. Moreover, as the ablation study on $\alpha$ is only conducted on two tasks (and not evaluated extensively), the influence of this parameter can possibly be even more substantial. While I can see tuning $\alpha$ might be possible in some cases and understand what the authors try to say by ``Figure 6 demonstrates that our approach maintains relatively few constraint violations even when the conservatism mechanism is removed (i.e., $\alpha=1$),” I do feel that the statement can be an over-claim.

---

> ### Author Response · Authors · 2024-08-13
>
> Apologies for the confusion. We would like to supplement the related explanations about W2 and W3.
>
> **W2: Justification of TD-reward surrogate**
>
> We will use an example to illustrate the effectiveness of the TD-error approximation. Let's assume there are three types of transitions: $(s,a,r_1,s_1),(s,a,r_2,s_2),(s,a,r_3,s_3)$, with the proportions $p$, $1-p$ and $0$ in the dataset, respectively.
>
> For the out-of-distribution transition $(s,a,r_3,s_3)$, it is expected to associate with high uncertainty and a large TD-error, according to previous research [1,2,3] and our empirical results in Figure 14.
>
> For the in-distribution transitions $(s,a,r_1,s_1),(s,a,r_2,s_2)$, the Q-value can be expressed as $Q(s,a)=p\cdot (r_1+V(s_1))+(1-p)\cdot(r_2+V(s_2))$. Thus, $(s,a,r_1,s_1)$ corresponds to a low TD-error $||Q(s,a)-r_1-V(s_1)||^2_2$ when it has a high probability of occurring in the dataset (i.e., when $p$ is close to $1$).
>
> Therefore, we expect to obtain $TD(s,a,r_1,s_1)<TD(s,a,r_2,s_2)<TD(s,a,r_3,s_3)$ when $p$ is high. In this case, TD reward aligns with the estimated dynamics $\hat P(s',r'|s,a)$ in the posterior for in-distribution transitions, where a higher occurrence probability leads to a higher TD reward. For out-of-distribution transitions, while the TD reward might not align with the estimated dynamics, it remains low to mitigate the adverse effects of inaccuracies in dynamics estimation. Similarly, for state distribution, a higher occurrence probability of states corresponds to a more accurate value function estimate and a higher TD reward, whereas out-of-distribution states typically result in a smaller TD reward.
>
> I hope this example provides an intuitive insight into this approximation. Rigorous theoretical derivations would involve the generalization of Q-value neural networks and their relationship to dynamics uncertainty, which are relatively complex and challenging. We would leave further theoretical analysis to our future work. Another potential direction is to adopt alternative approaches, such as explicit dynamic modeling, which could offer more straightforward theoretical results.
>
> [1] Bai, Chenjia, et al. "Pessimistic Bootstrapping for Uncertainty-Driven Offline Reinforcement Learning." ICLR 2021.
>
> [2] Rigter, Marc, Bruno Lacerda, and Nick Hawes. "One risk to rule them all: A risk-sensitive perspective on model-based offline reinforcement learning." NIPS 2024.
>
> [3] Ghosh, D., Ajay, A., Agrawal, P., & Levine, S.. Offline rl policies should be trained to be adaptive. ICML 2022.
>
>
>
> **W3: Conservative parameter $\alpha$**
>
> Our statement that "a roughly designed $\alpha$ will not significantly undermine our approach" was based on comparisons with the baselines, where our approach showed relatively fewer violations and a higher correlation between cost and reward (indicating the high policy diversity we expected) than baselines under various $\alpha$. We know this isn't a direct comparation shown in Figure 6 and apologize for any confusion caused by some of our claims in the response.
>
> We acknowledge that the influence of this parameter on the final performance cannot be overlooked, and we appreciate the reviewer for highlighting this. We believe that fine-tuning this parameter using a limited iteration of online interaction and re-adaptation could indeed address this issue. Additionally, the tasks used in the ablation experiments were randomly selected, and we plan to conduct these experiments across a broader set of tasks and report the results in a future version.
>
>
>
>
>
> Thank you again for the constructive feedback. We are happy to provide further explanations if there are still any confusions.

---

### Official Review · Reviewer_FywX · 2024-07-29

**Soundness:** 3
**Presentation:** 3
**Contribution:** 3
**Rating:** 7
**Confidence:** 3

**Summary:**

The paper tackles the important problem of adapting multi-objective policies to diverse preferences and constraints at deployment. This work uses state-of-the-art multi-objective RL methods to learn a set of policies that can align with diverse preferences. At deployment time, it doesn't assume access to the explicit hand-crafted preferences but introduces an adaptation method that uses demos to infer the target preferences. Additionally, the authors formulate this method to take into account explicit safety constraints at deployment time, inferred using only the demonstrations, without any online interaction. The approach is tested extensively on standard multi-objective and safe RL benchmarks, and they show that it can achieve higher performance, incur lower costs, and generate more diverse policies in these environments.

**Strengths:**

1. The paper tackles an important problem of adaptation in multi-objective and safe RL. The intuition of the overall method is well understood, the framework and the parameters are mentioned, and the results are shown over the standard environments.
2. The paper is well organized and presents a clear background of the past work in multi-objective RL. However, it would be nice to see a clearer distinction between the past approaches (PEDA etc.) used and changes (if any) made by the authors in this paper, for phase 1 of the methodology.
3. Past work assumed access to hand-crafted preferences or online interactions to infer the preferences at deployment time. However, this work bypasses that by using a Bayesian framework to infer the preference distribution at deployment time using a set of demos explaining the preferred behavior. This is a more practical and scalable approach to adapting policies at test time.
4. This method introduces an adaptation framework to infer the distribution of preferences from demonstrations. The authors lay out the theoretical and practical details for this step, including replacing the intractable Bayesian inference with a TD-error-based approximation from past literature.
5. The introduced method is generalized to include safety constraints in the RL agent. This transforms the constrained RL problem, into an unconstrained multi-objective RL setting, which is compatible with the methods introduced in this work. Further, to ensure that the given approximation doesn't lead to excessive safety violations the authors use a CVaR-based objective, to always overestimate the constraints in case of uncertainty due to low demos.
6. The authors demonstrate the effectiveness of their approach in several multi-objective and constrained simulation environments. The paper provides ablations to compare the effect of some hyperparameters and training methods on the efficacy of the method.

**Weaknesses:**

1. At training time, the method assumes access to the explicit preferences and demonstrations responding to those preferences. However, in practical settings, this information is not available even at training time, and the preferences and their clusters have to be learned in an unsupervised / self-supervised setting. I would be curious to know what the authors think about the effectiveness of this approach to learn the preferences and the policies in an end2end framework.
2. I would recommend the authors to present a clear distinction between the past work, and the approach finally used in this work to generate the multi-objective policies. In this current presentation, it is slightly confusing to understand the particular contributions/adaptations of past work  to incorporate this with the posterior approximation framework. A pseudocode explaining the entire algorithm would make the method clearer to the readers.
3. In the background, it would be nice to see some discussion contrasting the work in multi-objective adaptation to inverse RL, because in the introduction the authors mention - “it is more natural to infer expected behaviors through a few demos that implicitly indicate….” Comparing this to the approaches in inverse RL, and further, in Bayesian RL would be very useful to highlight the contributions.
4. Under expert demos, the prompt-MODT baseline performs comparably to the methods introduced in this work. I would recommend the authors provide a wider discussion of the effectiveness of explicitly modeling human preferences using the adaptation module and if using the distributional Bayesian approach makes it robust to noisy samples.

**Questions:**

1. The method is based on settings where the total reward is a linear combination of preferences and individual rewards. How can this setting be adapted to settings where we have access only to the total reward and demos and not the individual components?
2. In the evaluation metrics, I would be curious to know why is the hypervolume and therefore, the diversity in the policies important. For the baselines, we observe that the policies with low hypervolume still have considerably close performance to this work in terms of utility and cost. A paragraph along these lines would allow a broader audience to understand the significance of these experiments and results.
3. In certain environments, despite violating constraints, the returns are high, which indicates that the cost and the rewards are independent of each other. This is slightly confusing and I would appreciate some clarification in this respect.
4. In Figure 6, it would be nice to analyze why higher demos incur a higher cost.

**Limitations:**

1. The method assumes access to the preferences at training time, which might not be practical for real-world settings, where the individual objectives and the preference distribution are unknown. This could be the reason that the method outperforms the prompt-MODT baseline i.e. the error introduced due to implicitly modeling the preferences, and not having access to the target distribution could potentially decrease the effectiveness of PDOA.
2. The authors mention that the adaptation step is not real-time, which makes it difficult to transfer this system to real-world RL systems such as LLMs and robots.
3. The evaluation is limited to locomotion environments, so, I suggest the authors include a discussion on the implications of the method and evaluation towards adaptation and safety in real-world RL problems.

---

> ### Author Rebuttal · Authors · 2024-08-05
>
> We thank the reviewer for all the valuable comments. Please refer to the point-to-point responses below.
>
> **W1: Unavailable access to the explicit preferences**
>
> We agree that unavailable access to preference labels is indeed a problem in real-world MORL applications. Some unsupervised techniques, such as clustering methods or contrastive learning, might help address this issue. In our paper, we adopt a simple yet effective solution proposed by MODF, which approximates the preference label of each trajectory with its normalized return vector ($R/|R|$) (see Appendix A.5). In our safe RL and CMORL experiments, where preference labels are unavailable, this simple solution leads to satisfactory performance.
>
> Moreover, our approach has the potential to promote preference learning when real labels are unavailable. Specifically, one can use the roughly approximate preference labels to train preference-conditioned models and then incorporate them into our framework to infer more precise preference labels for unlabeled trajectories in the dataset.
>
>
>
> **W2: Distinction between past work and ours, and pseudocode**
>
> Briefly, this paper studies a new and more realistic MORL problem (i.e., generating the expected behaviors without access to target preference) that differs from previous works. Our main contribution is a flexible framework to solve this new problem, rather than a new MORL method paralleled to MODF and PEDA. Specifically, existing MORL methods can be easily incorporated into our framework once they provide preference-conditioned policies or values, enabling them to solve this new problem that the original methods cannot. In our paper, we use the original implementation of MODF and PEDA without change, except for removing the RWA component of MODF due to its online interactions.
>
> We will add pseudocode in a future version to improve understanding.
>
>
>
> **W3: Association with Inverse RL and Bayesian RL**
>
> We appreciate the reviewer highlighting these connections. Preference inference is similar to identifying the decisive rewards of demonstrations, which is also the goal of IRL. Unlike traditional IRL methods, our work assumes a certain reward information and focuses on identifying preferences during deployment rather than learning decisive rewards during training.
>
> Bayesian RL is fundamental to our theoretical derivation and explored in many RL sub-fields. While most studies focus on enhancing online exploration and managing uncertainty in model-based or offline RL, our approach employs Bayesian inference to address the uncertainty of the target preference distribution using a few demonstrations.
>
> We will include more detailed discussions in the future version.
>
>
>
> **W4: Comparison to prompt-based methods**
>
> First, explicit preference modeling allows us to incorporate safety considerations into the MORL to enhance safety, while prompt-based methods lack safety guarantees and perform poorly in safe RL tasks. Additionally, explicit preference modeling allows users to verify the inferred preferences, thereby improving reliability. Besides, since prompt-based methods are designed for multi-task RL, they require users to spend extra effort transforming MORL datasets into multiple task datasets (see Appendix A.5), whereas our approach doesn't. Moreover, high noise in samples can be reflected as high variance in the inferred distribution, which can then be utilized to promote constraint adherence.
>
>
>
> **Q1: Access to only demos' scalarized reward, not vector reward**
>
> Only the demos' scalarized utility may provides limited information for policy learning as utilities under different preferences can't be directly compared. This makes evaluating each trajectory's performance difficult, turning the problem into unsupervised RL. In this case, a potential solution is to cluster trajectories and imitate those most similar to the given demonstrations.
>
> **Q2: Importance of hypervolume and diversity as metrics**
>
> The utility metric in our paper measures average utility across target preferences. But it may not accurately reflect overall performance if reward scales vary greatly. In contrast, the hypervolume metric measures the volume enclosed by the learned policies' returns, directly reflecting Pareto front performance, which is crucial in MORL research. Hypervolume is also less sensitive to reward scaling and is a common metric in previous MORL studies, facilitating fair comparisons.
>
>
>
> **Q3: Relationship between return and cost**
>
> Cost and reward generally correlate positively, so high constraint violations (i.e., high costs) often come with high rewards. However, poorly learned policies can lead to both high costs and low rewards.
>
> **Q4: Connection between the number of demos and cost in Figure 6**
>
> More demonstrations reduce uncertainty (variance of adapted preference distribution) by lessening the entropy item's effect in Eq. (7), which then weakens the conservatism estimation (Eq. (10)). Thus, both cost and reward increase.
>
>
>
> **L1: Unavailable access to explicit preferences**
>
> Related discussions are provided in response to W1.
>
> **L2: Real-Time adaptation**
>
> We think that frequent preference switching isn't very common in real-world applications. Additionally, the time bottleneck is more likely the collection of new demonstrations rather than the inference. Besides, using fewer gradient update steps and simplifying the network structure could improve inference speed.
>
> **L3: Discussion on real-World RL problems**
>
> We detail the application of our approach to real-world RL problems in Appendix A.10. We emphasize that designing target preferences or safety thresholds is a complex task requiring extensive trial and error in real-world scenarios.
>
> Furthermore, previous MORL methods that could be part of our framework have been tested in various environments like discrete grid-world and game tasks. Therefore, we believe our approach is promising for diverse real-world tasks beyond locomotion environments.

---

> > ### Comment · Reviewer_FywX · 2024-08-11
> >
> > I thank the authors for their response. I have read the rebuttal and maintain my score.

---

### Author Rebuttal · Authors · 2024-08-06

We appreciate all the valuable comments from the reviewers. We have provided point-to-point responses to each reviewer's questions and will revise our manuscript according to their suggestions.

---

### Decision · Program_Chairs · 2024-09-25

**Decision:**

Accept (poster)

**Comment:**

The paper studies a well-motivated, significant problem (multi-objective reinforcement learning) and proposes an offline adaptation approach that does not require preferences between objectives to be indicated during deployment time (instead they are inferred from offline demonstrations).
The reviewers agreed the proposed approach is technically sound, novel, and the empirical evaluations and ablations convincingly show the merits of the proposal. During the reviewer-author discussion, the authors clarified several weaknesses pointed out by the reviewers; incorporating them into a revision will substantially strengthen the paper.